# C/N ratio, stable isotope ($\delta^{13}$C, $\delta^{15}$N), and *n*-alkane patterns of brown mosses along hydrological gradients of low-centred polygons of the Siberian Arctic

**Romy Zibulski[1,2], Felix Wesener[4], Heinz Wilkes[3,5], Birgit Plessen[3,], Luidmila A. Pestryakova[6], Ulrike Herzschuh[1,2,7]**

[1] {Alfred-Wegener-Institut Helmholtz-Zentrum für Polar- und Meeresforschung, 14473 Potsdam, Germany}

[2] {University of Potsdam, Institute of Earth and Environmental Sciences, 14476 Potsdam-Golm, Germany}

[3] {Helmholtz Centre Potsdam GFZ German Research Centre for Geosciences, 14473 Potsdam, Germany}

[4] {Humboldt University of Berlin, Institute of Biology, 10115 Berlin, Germany}

[5] {Institute for Chemistry and Biology of the Marine Environment (ICBM), Carl von Ossietzky University, 26111 Oldenburg, Germany}

[6] {Northeast Federal University, Department for Geography and Biology, 677000 Yakutsk, Russia}

[7] {University of Potsdam, Institute of Biochemistry and Biology, 14476 Potsdam-Golm, Germany}

Correspondence to: Romy Zibulski (R. Zibulski@awi.de)

## Abstract

Mosses are a major component of the arctic vegetation, particularly in wetlands. We present C/N ratiosmolar, $\delta^{13}$C and $\delta^{15}$N data of 400 brown moss samples belonging to 10 species that were collected along hydrological gradients within polygonal mires located on the southern Taymyr Peninsula and the Lena River delta in northern Siberia. Additionally, *n*-alkane patterns of six of these species (16 samples) were investigated. The aim of the study is to see whether the inter- and intraspecific differences in C/N, isotopic compositions and *n*-alkanes are indicative of habitat with particular respect to water-level. Overall, we find high variability in all investigated parameters for two different moisture-related groups of moss species. The C/N$_{(m)}$ ratios range between 11 and 53 (median: 32) and show large variations at the intraspecific level. However, species preferring a dry

habitat (xero-mesophilic mosses) show higher $C/N_{(m)}$ ratios than those preferring a wet habitat (meso-hygrophilic mosses). The $\delta^{13}C$ values range between -37.0 and -22.5‰ (median = -27.8‰). The $\delta^{15}N$ values range between -6.6 and +1.7‰ (median = -2.2‰).We find differences in $\delta^{13}C$ and $\delta^{15}N$ compositions between both habitat types. For some species of the meso-hygrophilic group, we suggest that a relationship between the individual habitat water-level and isotopic composition can be inferred as a function of microbial symbiosis. The *n*-alkane distribution also shows differences primarily between xero-mesophilic and meso-hygrophilic mosses, i.e. having a dominance of *n*-alkanes with long (*n*-$C_{29}$, *n*-$C_{31}$) and intermediate (*n*-$C_{25}$) chain lengths, respectively. Overall, our results reveal that $C/N_{(m)}$ ratios, isotopic signals and *n*-alkanes of studied brown moss taxa from polygonal wetlands are characteristic of their habitat.

# 1. Introduction

Specific physiological and morphological traits enable mosses to attain extensive surface coverage in low-temperature ecosystems such as arctic tundra (Turetsky et al., 2012; Wasley et al., 2006). Mosses are a major component of the biomass in arctic wetlands and contribute strongly to the biodiversity. In particular, the vegetation of the widely distributed polygonal tundra is rich in moss taxa, which partly originates from the strong compositional turnover along a small-scale hydrologic gradient (Zibulski et al. 2016). As an intermediate layer between air and the permafrost soil, mosses control the water cycle, greenhouse gas and energy exchange (Blok et al., 2011; McFadden et al., 2003), and the structure of the habitats of vascular plant communities in arctic landscapes (Beringer et al., 2001; Gornall et al., 2011; Malmer et al., 1994). Because of their low decomposition rates (Aerts et al., 1999; Turetsky, 2003), they contribute strongly to the peat and permafrost carbon pool (Tarnocai et al., 2009).

Despite the significance of mosses in high-latitude biodiversity and matter cycles only little is known about their C/N ratio, stable isotope and *n*-alkane characteristics in comparison to vascular plants. Such information is not only necessary to improve our understanding of the physiological adaptation or plasticity of mosses to certain environmental characteristics, but can also be used when

similar measurements of fossil plant material are interpreted as proxies of former ecological or
environmental conditions (e.g. Birks, 1982).
Carbon-to-nitrogen ratios (C/N), stable carbon and nitrogen isotope values ($\delta^{13}$C, $\delta^{15}$N) and the
*n*-alkane fingerprints of bulk organic matter are among the most commonly measured parameters of
plant matter (Chambers and Charman, 2004). The C/N ratios of mosses are generally in the range of
those of higher terrestrial plants (Prahl et al., 1980) but are known to depend on the available nitrogen,
which originates in pristine regions from decomposition of organic matter, microbial activity or
atmosphere deposits (e.g. Chanway et al., 2014; Lee et al., 2009). We provide C/N ratios by weight of
arctic brown mosses, in anticipation that they will be useful for comparative palaeo-environmental
reconstructions (Andersson et al. 2011, ) and in the evaluation of organic matter sources in Russian
permafrost soils with regard to species and habitat-specific patterns. Furthermore, C/N ratios may be
related to growth form, i.e. higher ratios can be expected for mosses which compete with vascular
plants for light and thus need to invest in a high stem-stability (Sveinbjörnsson and Oechel, 1992).

69        As well as the C/N ratio, the $\delta^{13}$C composition is barely altered by decomposition processes in the

soil and also suitable for palaeo-environmental reconstructions. The $\delta^{13}$C ratio of an individual plant
are a mixed signal of the physiological traits of the species and the individual's direct environment.
Mosses use the $C_3$ pathway for carbon assimilation (O'Leary, 1988, Farquhar et al., 1989) and despite
a lack of stomata in the photosynthetic active parts, they have a similar range in their $\delta^{13}$C values of
between -24 and -32 ‰ (Ménot and Burns, 2001; Rundel et al., 1979; Smith and Epstein, 1971) as
vascular $C_3$ plants. Differences in $\delta^{13}$C values among several species can be explained by individual
plant physiology and biochemistry (Galimov, 2000). Differences within a single species have been
linked with environmental conditions such as temperature (Skrzypek et al., 2007; Waite and Sack,
2011), growing depth below water-level (Raghoebarsing et al., 2005), position within a cushion or
hummock (Price et al., 1997), lipid content (Rundel et al., 1979) or the influence of microbial
symbioses (Liebner et al., 2011; Vile et al., 2014). Furthermore, even differences among branches and
stems of single individuals have been reported (Loader et al., 2007). However, almost all of these
studies were made on Sphagnaceae, which are not representative of all mosses because of their
specific morphology (i.e. the occurrence of a photosynthetic active cell type and a dead cell type,
which is responsible for water storage and often an additional coating of the photosynthetic active cell)
and their specific habitat preferences (i.e. preferring acidic wetlands). Hence, this information cannot
simply be transferred to brown mosses – which form the major component in northern Siberian
lowlands – due to their different morphology.
Most studies on $\delta^{15}$N values of moss material have focused on the anthropogenic impact on the
nitrogen cycle (Harmens et al., 2011; Liu et al., 2008; Poikolainen et al., 2009), whereas reports on the
$\delta^{15}$N compositions of mosses from relatively pristine ecosystems such as the Arctic are rare or have
been investigated in relation to the study of bird colonies (e.g. Lee et al., 2009). Potentially, such
information can indicate pathways and sources of the nutrient supply in these N-limited ecosystems
(Kielland, 1997; Michelsen et al., 1996, 1998).
Compared with vascular plants and *Sphagnum* species, relatively few investigations of *n*-alkane
patterns of brown mosses are available. Palaeoenvironmental reconstructions use the potential of
*n*-alkanes to distinguish between different plant groups (Ficken et al., 1998, moisture conditions
(Pancost et al., 2000, Nichols et al 2006, Zhou et al. 2010), whether organic material is to decomposed
and changes in temperature (Feakins et al., 2016, Sachse et al., 2006) along distinct gradients.
*Sphagnum* species, for example, show a dominance of *n*-$C_{23}$ and *n*-$C_{25}$ homologues (Baas et al., 2000)
that are comparable to the pattern of vascular submerged plants (Ficken et al., 2000), and ratios are
used as a proxy for wet moisture conditions. Other studies discuss the suitability of *n*-alkane patterns
in moss species for chemotaxonomical studies on recent and fossil material (Bush and McInerney,
2013; Nott et al., 2000; Schellekens and Buurman, 2011). A greater protection potential of waxes with
a higher content of long-chain *n*-alkanes against solar irradiation or alternatively an enhanced loss of
short-chain *n*-alkanes by evaporation have been suggested as possible mechanisms to cause different
*n*-alkane patterns in leaf waxes of trees (Sachse et al., 2006). However, with respect to mosses the
pattern and mechanisms are even less understood.
This study presents C/N$_{(m)}$, isotopic ($\delta^{13}$C, $\delta^{15}$N) and *n*-alkane characteristics of mosses from low-
centred polygons in northern Siberia. Low-centred polygons are geomorphological forms in arctic
landscapes originating from frost-heave processes in the soil. They are characterized by elevated dry
rims and a water-saturated or water-filled centre. This centimetre-scale hydrological gradient is well
reflected by a strong turnover in the vascular plant and moss composition (Zibulski et al., 2016). We
investigate the relationship between the $C/N_{(m)}$, isotopic and *n*-alkane compositions and the
hydrological conditions within low-centred polygons. In particular, we aim to reveal whether
fingerprints are habitat-specific or rather species-specific.

## 2. Material and methods

### 2.1 Sites

The plant material was collected during the vegetation season (July-August) from eight low-centred
polygons located along a zonal vegetation gradient ranging from open forest via the forest-tundra
intersection to subarctic tundra (Matveev, 1989) to obtain a representative sample set of northern
Siberian lowlands (Fig. 1). Six of the polygons (06/P, 17/P, P3/I, P3/II, P3/III, 12/P), sampled in 2011,
are located in the Khatanga River region (70–72° N, 97–102° E, southern Taymyr Peninsula) and a
further two polygons (LP1 and LP2), sampled in 2012, are located on Samoylov Island in the Lena
River Delta (72.4° N, 126.5° E). The climate is cold-humid (Khatanga climate station annual mean
temperature and precipitation: -13.2°C and 272 mm, Rivas-Martinez and Rivas-Saenz, 2009;
Samoylov climate station annual mean temperature and precipitation: -12.5°C and 232.7 mm, Boike et
al., 2013)).

### 2.2 Sampling and studied moss species

A representative continuous transect of adjacent one square-metre plots (from rim to rim) was selected
for sampling in each polygon. The surface height in relation to water-level (measured at the centre of
each plot) and vegetation (abundance information) were recorded. Individual characteristics of each
low-centred polygon are presented in Table A1. Moss plants were hand-picked, dried in the field and
transported to the Alfred Wegener Institute in Potsdam. Taxa identification is based on the relevant
literature (Frahm and Frey, 2004; Lobin et al., 1995; Smith, 1978) and the Bryophyte Flora of North
America internet source (http://www.efloras.org).
In total, eight species that were observed to have different water-regime preferences were included in
the analyses. To approximate water-regime preferences of each species, we calculated the mean plant-
position in relation to water-level (h) from the recorded plot surface height. *Warnstorfia exannulata*
was observed to grow as part of a swinging mat at water-level, it was assigned a preference for water-
level. Using this information the species were classified as either as of xero-mesophilic mosses (mean
plant-position between 13 and 16 cm in relation to water-level) or to the group of meso-hygrophilic
mosses (mean plant-position between 3 and -30 cm in relation to water-level) to ease the presentation
of results.
**Table 1:** *Calculated mean plant-position relative to the water-level and the classification of the sampled brown moss species.*
*(\* Samples of* Warnstorfia exannulata *were growing on a swinging mat at water-level and mean plant position was thus set to*
*zero.)*

| species | abbreviation | h (mean plant-position in relation to water-level in cm) | classification |
|---|---|---|---|
| *Hylocomium splendens* | Hyl_spl | 16 | xero-mesophilic |
| *Tomentypnum nitens* | Tom_nit | 13 | xero-mesophilic |
| *Aulacomnium turgidum* | Aul_tur | 13 | xero-mesophilic |
| *Aulacomnium palustre* | Aul_pal | 13 | xero-mesophilic |
| *Hamatocaulis lapponi* | Ham_lap | 3 | meso-hygrophilic |
| *Warnstorfia exannulata* | War_exa | 0 * | meso-hygrophilic |
| *Meesia triquetra* | Mee_tri | -1 | meso-hygrophilic |
| *Drepanocladus revolvens* | Dre_rev | -5 | meso-hygrophilic |
| *Scorpidium scorpioides* | Sco_sco | -13 | meso-hygrophilic |
| *Calliergon giganteum* | Cal_gig | -30 | meso-hygrophilic |


## 2.4 Measurements of C/N$_{(m)}$ ratios, stable carbon and nitrogen isotope values and *n*-alkane distributions

Selected plant material (i.e. apical parts of a specimen) was rinsed with de-ionised water and
mechanically cleaned from organic particulate material. The content of carbon and nitrogen and the
ratio of stable isotopes were measured with a DELTAplusXL isotope ratio mass spectrometer (Thermo
Fischer Scientific) coupled to an elemental analyser (NC2500 Carlo Erba) via a CONFLOW III
Interface. Due to the relatively wide range of C/N ratios of mosses, we used about 1.5 mg for each
carbon stable isotope measurement ($n = 400$) and a replicate of 3 mg for each nitrogen stable isotope
measurement ($n = 326$) and the analysis of elemental composition. The high weight needed for the
nitrogen sample replicates prevented the measurement of $\delta^{15}N$ and thus the C/N$_{(m)}$ for some samples.
The calibration for carbon was performed using an urea standard and a $\delta^{13}C$ isotopic standard (IAEA
CH-7). The nitrogen contents were calibrated against an acetanilide standard and the nitrogen isotopic
composition with ammonium sulphate standard (IAEA N-1). The reliability of the method was
checked with the NIST plant standard SRM 1547. The isotopic ratios are given in delta notation

relative to VPDB for $\delta^{13}$C values and relative to air for $\delta^{15}$N values, respectively. The reproducibility for replicate analyses is 0.2% for carbon and nitrogen and 0.2‰ for $\delta^{13}$C and $\delta^{15}$N values.

*n*-Alkane analyses were performed on a subset of 16 samples. We took material from the polygon complex P3 from the Khatanga region (72.149° N, 102.693° E), which comprises three successive low-centred polygons (Table A1) to exclude effects of environmental conditions at different locations. The moss samples were washed, identified and air-dried. They were weighed (150–1000 mg dry weight) and samples extracted with an accelerated solvent extractor (ASE) (Dionex, Sunnyvale; USA) using $Cl_2$Me:MeOH (10:1) at 5 bar and 75°C. The extract was separated in to aliphatic hydrocarbon, aromatic hydrocarbon and nitrogen, sulphur and oxygen compound fractions using medium pressure liquid chromatography following Radke et al. (1980). Five µg of the quantification standard (5α-androstane, 1-ethylpyrene, 5 α-androstan-17-one and erucic acid) were added . Gas chromatography (GC) of aliphatic hydrocarbon fractions was performed using a GC Agilent 6890 equipped with an Ultra-1 fused silica capillary column (Model Agilent 19091A-105, length 50 m, inner diameter 200 µm, film thickness 0.33 µm). Helium was used as a carrier gas at a constant flow rate of 1 ml $min^{-1}$. The GC oven was heated from 40 °C (2 min hold time) to 300 °C (65 min hold time) at a rate of 5°C per minute. The samples were injected by means of splitless injection. Quantitative evaluation of data was done with ChemStation software. Additionally, we calculated the average chain length ($ACL_{21-33}$) assumed to represent a proxy for moisture (Andersson et al, 2011) and temperature (Bush and McInerney, 2015) with a comprehensible extended range from $n$-$C_{21}$ to $n$-$C_{33}$ and the proxy ratio $P_{aq}$, which was developed as a proxy ratio to distinguish submerged or floating aquatic macrophytes from emergent and terrestrial plants (Ficken et al., 2000):

$$ACL_n = \frac{\sum(n \times C_n)}{\sum C_n} \quad , n = 21 - 33$$

$$P_{aq} = \frac{(C_{23} + C_{25})}{(C_{23} + C_{25} + C_{29} + C_{31})}$$

## 2.5 Statistical Tests

We calculated the range, .25, .5, and .75 quantiles of $\delta^{13}C$ and $\delta^{15}N$ values and the $C/N_{(m)}$ ratios for all
species (Table A2). Significant differences in the $C/N_{(m)}$ ratios and $\delta^{13}C$ and $\delta^{15}N$ values among
different groups were assessed with a t-test. In addition, we performed linear regression between the
obtained values for each individual species and surface height. All analyses were implemented in R
version 3.2.0 (R Core Team, 2015). Furthermore, we performed a PCA with the percentage of
*n*-alkane homologues (square-root transformed) in R using the vegan package version 2.0-10
(Oksanen et al., 2013).

## 3. Results

The overall ranges in $C/N_{(m)}$ ratios of both groups have a broad overlap (xero-mesophilic: 22.5–67.9;
meso-hygrophilic: 15.4–70.4). However, the medians of the $C/N_{(m)}$ values of the xero-mesophilic
species ranging from 47.6 to 52.9 (Fig. 2) are significantly higher than those of the meso-hygrophilic
group, which range from 37.1 to 46.5 (W = 18280, $p \ll 0.001$). The $C/N_{(m)}$ ratios show no
intraspecific relations among individual species and water-level (Fig. 3a), except for *Tomentypnum*
*nitens* ($r^2 = 0.11$, $p < 0.05$).
The $\delta^{13}C$ values of the meso-hygrophilic group (-34.9 to -22.5‰) cover the range of the xero-
mesophilic group (-32.1 to -24.2‰), which have a noticeably lower variance in $\delta^{13}C$ values (Fig. 2).
The medians of the individual species in the xero-mesophilic group (range: -29.4 to -27.1‰) are
significantly different (W = 16232, $p = 0.008$) from those of the meso-hygrophilic group (range: -29.2
to -24.8‰). $\delta^{13}C$ values of *Meesia triquetra, Drepanocladus revolvens,* and *Scorpidium scorpioides*
(all belonging to the meso-hydrophilic group) are significantly positively related with the position of
the water-level (Fig. 3b), while no member of the xero-mesophilic group revealed such relationship
(Fig. 3a).
The ranges of $\delta^{15}N$ values of both groups are rather similar (Fig. 2). However, individual species
medians of both groups are significantly different (t = -6.96, $p \ll 0.001$; xero-mesophilic group; -3.2
to -2.7‰, meso-hygrophilic group: -2.9 to -0.1). *Drepanocladus revolvens, Scorpidium scorpioides,*
and *Calliergon giganteum*, all belonging to the meso-hygrophilic group, exhibit a positive relationship
between the $\delta^{15}N$ values and position relative to water-level (Fig. 3b.
*n*-Alkane distributions show the expected predominance of the odd chain length (Table 2). The
absolute *n*-alkane concentrations ($n$-$C_{19}$ to $n$-$C_{33}$) range from 34 to 238 µg g$^{-1}$ of dry weight. The
individual taxa show significant differences with respect to carbon number of the most abundant
*n*-alkane ($n$-$C_{max}$) forming unimodal distribution patterns. It is $n$-$C_{31}$ for *Tomentypnum nitens*, which is
also the only species containing $n$-$C_{33}$ in detectable amounts, $n$-$C_{29}$ for *Aulacomnium turgidum*, $n$-$C_{27}$
for *Aulacomnium palustre*, $n$-$C_{27}$ and $n$-$C_{25}$ for the *Drepanocladus* taxa, and $n$-$C_{max} = 25$ for
*Scorpidium scorpioides.* These differences in the *n*-alkane composition largely relate with the species-
preferred mean plant-position relative to water-level (Fig. 4).
Evaluations of the *n*-alkane biomarker proxies, $ACL_{21-33}$ and $P_{aq}$, also show a clear division
between the xero-mesophilic and the meso-hygrophilic species groups (Table 2), whereas intraspecific
variations are rather small (with the exception of *Drepanocladus*). The xero-mesophilic group is
notable for high averages of $ACL_{21-33}$ (28.41) and $P_{aq}$ (0.17) compared to low averages of $ACL_{21-33}$
(25.61) and $P_{aq}$ (0.87) for the meso-hygrophilic group.
The observed trend is also assumed in the biplot of the first two PCA axes, even though their
explained variance is relatively low (16.9%) in the dataset (Fig. 5). The first axis separates xero-
mesophilic from meso-hygrophilic taxa. *Aulacomnium* taxa are located in the upper range and
*Tomentypnum nitens* in the lower range of the second PCA axis, while no trend is observed within the
meso-hygrophilic group along the second axes.
**Table 2:** *The concentration (µg/g dry weight) and $ACL_{21-33}$ and $P_{aq}$ (after Ficken et al., 2000) of* n-*alkanes in*
*brown moss samples. (Numbers in brackets relates to the sample number.)*

| species | $n$-$C_{19}$ | $n$-$C_{21}$ | $n$-$C_{23}$ | $n$-$C_{25}$ | $n$-$C_{27}$ | $n$-$C_{29}$ | $n$-$C_{31}$ | $n$-$C_{33}$ | $ACL_{21-33}$ | $P_{aq}$ |
|---|---|---|---|---|---|---|---|---|---|---|
| **Xero-mesophilic habitat group** | | | | | | | | | | |
| Tom_nit (1) | 0 | 2.0689 | 2.6374 | 6.0816 | 10.7454 | 17.7273 | 34.0742 | 0 | 28.86 | 0.14 |
| Tom_nit (2) | 0.5571 | 1.041 | 1.4493 | 2.6964 | 9.1008 | 8.4867 | 20.2818 | 7.2312 | 29.47 | 0.13 |
| Tom_nit (3) | 0.9936 | 1.456 | 2.175 | 2.8712 | 9.6118 | 7.2721 | 21.2309 | 4.9717 | 29.14 | 0.15 |
| Aul_tur (1) | 0 | 0.968 | 1.126 | 2.2001 | 8.9548 | 23.5267 | 3.5595 | 0 | 28.15 | 0.11 |
| Aul_tur (2) | 1.028 | 1.3468 | 1.2794 | 4.8797 | 18.7427 | 50.9291 | 5.5645 | 0 | 28.22 | 0.10 |
| Aul_pal (1) | 0 | 1.9298 | 2.5459 | 4.7646 | 20.5085 | 7.5826 | 7.979 | 0 | 27.35 | 0.32 |
| Aul_pal (2) | 0.7341 | 1.0767 | 1.5183 | 2.9728 | 12.3293 | 11.0909 | 4.9109 | 0 | 27.69 | 0.22 |

**Meso-hygrophilic habitat group**

| Dre_rev (1) | 0.7868 | 1.5426 | 2.3659 | 43.833 | 19.0836 | 5.1551 | 3.5637 | 0 | 25.92 | 0.84 |
|---|---|---|---|---|---|---|---|---|---|---|
| Dre_rev(2) | 0.981 | 1.5227 | 2.7605 | 22.5638 | 14.5103 | 6.5356 | 7.5623 | 0 | 26.6 | 0.640 |
| Dre_sp. | 1.4696 | 1.9968 | 5.0986 | 29.6729 | 30.4582 | 9.9108 | 5.1317 | 0 | 26.38 | 0.70 |
| Sco_Sco (1) | 0 | 3.7612 | 11.7002 | 133.4207 | 29.0024 | 6.2023 | 1.9425 | 0 | 25.3 | 0.95 |
| Sco_Sco (2) | 0 | 3.8911 | 10.4693 | 93.7009 | 21.4601 | 5.8531 | 1.7706 | 0 | 25.29 | 0.93 |
| Sco_Sco (3) | 0.8856 | 2.7949 | 11.8988 | 134.378 | 24.2348 | 5.4035 | 2.4969 | 0 | 25.28 | 0.95 |
| Sco_Sco (4) | 1.6217 | 5.5813 | 11.705 | 101.7602 | 21.3126 | 6.9161 | 2.4522 | 0 | 25.26 | 0.92 |
| Sco_Sco (5) | 1.4083 | 3.8857 | 11.8966 | 121.5701 | 23.3167 | 4.315 | 1.7261 | 0 | 25.21 | 0.96 |
| Sco_Sco (6) | 1.345 | 2.6672 | 19.4796 | 170.3015 | 34.3255 | 6.8599 | 3.1732 | 0 | 25.28 | 0.95 |

# 4. Discussion

## 4.1 $C/N_{(m)}$ ratios

The C/N ratios of mosses from polygonal tundra in Northern Siberia are relatively low compared with those obtained for mosses from Antarctic bogs that range between 80 and 100 (Björck et al., 1991) or from western Canada that range between 55 and 76 (Kuhry and Vitt, 1996). However, neither the taxa xeric and mesic growing conditions were sampled in Antarctica and Canada. All investigated species are considered as ectohydric mosses, which receive nitrogen mostly from precipitation deposits (Ayres et al., 2006). Our results reveal that averaged $C/N_{(m)}$ ratios for the xero-mesophilic moss group are higher than for the meso-hygrophilic group, probably reflecting the known difference between terrestrial and aquatic plants (Meyers and Ishiwatari, 1993). There are two possible impacts, which can influence the C/N ratio of these groups: (1) competition with vascular plants and (2) accessibility of nitrogen pools (2)g. (1) If moss plants invest in a high stem-to-leaf biomass ratio, which results in a high $C/N_{(m)}$ ratio, they will increase their height and stability, and thus their competitive ability against vascular plants for light (Sveinbjörnsson and Oechel, 1992). Furthermore, the low N input by precipitation and a low N content of moss litter slows down the fungal and bacterial N mineralisation which increases the thickness of moss litter mats (Gornall et al., 2007; Turetsky, 2003). This in turn will increase the isolating function of moss mats, thus negatively affecting seed germination of vascular plants (Gornall et al., 2007). (2) Lower $C/N_{(m)}$ ratios of meso-hygrophilic mosses may originate from higher amounts of dissolved nitrogen in polygon waters as a result of high net primary productivity, the presence of $N_2$-fixers such as cyanobacteria, and the exudations of zooplankton. Frahm (2001) assumes that loose epiphytic and endophytic symbiotic relationships between mosses

and cyanobacteria are probably restricted to wetland taxa. Lindo et al. (2013) report such associations
between brown mosses and cyanobacteria. Thus, the N supply is better for brown mosses preferring
meso-hygrophilic than xero-mesophilic habitats, and the respective taxa accordingly have lower
$C/N_{(m)}$ ratios indicating habitat-specific variation in $C/N_{(m)}$ ratios. We expected to also find
intraspecific variations between $C/N_{(m)}$ ratios and water-level. The large variability in the C/N data
may be a result of atmospheric conditions and organic matter degradation being the principal sources
at xeric sites, whereas in mesic and wet sites microbial symbionts play an important role in the C/N
ratio. However, the signal-to-noise ratio is probably too low to give a meaningful result because only
the average water level of each plot but not of each individual plant was recorded.

## 4.2 $\delta^{13}C$ values

With respect to bryophytes, most isotopic studies have hitherto been performed on *Sphagnum* (Markel
et al., 2010, Ménot and Burns, 2001) while our study focuses on brown mosses – a major component
in Siberian wetlands. The intraspecific variability for some meso-hygrophilic species (i.e. *Meesia*
*triquetra, Drepanocladus revolvens, Scorpidium scorpioides*) show that the $\delta^{13}C$ signals are related to
the hydrological conditions at the growing site of each individual, i.e. individuals growing at dry sites
showed higher medial $\delta^{13}C$ values than those growing at wet sites. A difference among the two habitat
groups is observed; they partly contradict the intraspecific findings in that some of the xero-mesophilic
species known to prefer dry rims such as *Hylocomium splendens* and *Tomentypnum nitens* have
particularly low $\delta^{13}C$ medians.
The detected differences in moss $\delta^{13}C$ values, particularly of the meso-hygrophilic group, either
reflect a source signal depending on water-level or a physiological reaction of the plant related to
water-level (Bramley-Alves et al., 2014; Proctor et al., 1992). Mosses are typical $C_3$ plants (Farquhar
et al., 1989, Rundel et al., 1979) characterized by a high $CO_2$ compensation point (Bain and Proctor,
1980; Dilks and Proctor, 1975; Salvucci and Bowes, 1981). The high availability of atmospheric $CO_2$
and elevated diffusion rates of $CO_2$ in air compared to water (O'Leary) result in typical terrestrial $C_3$
land plant $\delta^{13}C$ characteristics, because of a decreasing cell water pressure in dry habitats which entails
a strong discrimination rate against $^{13}CO_2$ induced by RuBisCO (Rice and Giles, 1996). With respect
to the xero-mesophilic group, we observe an increase in discrimination against $^{13}C$ from taxa
preferring a low position relative to the water-level (e.g. *Aulacomnium* taxa) than those preferring high
positions (e.g. *Hylocomium splendens*). In contrast, if plant tissue is coated by a water film, the cell
water pressure should reach an optimum, which is expected to results in a weaker discrimination rate
against $^{13}C$ by RuBisCO (Rice and Giles, 1996), because of a source restriction by the slower diffusion
rate of $CO_2$ in water (Lloyd and Farquhar, 1994). A lower carbon isotope discrimination related to
water saturation is observed for only three species out of six meso-hygrophilic mosses. However, this
basic signal may be masked by variations in $\delta^{13}C$ values of different carbon sources, which are
expected to be more influential for meso-hygrophilic mosses in water-saturated conditions. Ménot and
Burns (2001) studied intraspecific variations for three *Sphagnum* species, which prefer three different
habitat types (dry, meso, wet) along an elevational gradient, which was positively correlated with
precipitation. They find a decline in discrimination against $^{13}C$ with increasing wetness, and similar to
our results no relationship for species with a strong wet –preference. This is attributed to the variation
in $\delta^{13}C$ from highly varying dissolved inorganic carbon (Proctor et al., 1992). Mosses potentially
access $^{13}C$-depleted $CO_2$ that originates from oxidation of typically strongly $^{13}C$-depleted biogenic
methane by methanotrophic microorganisms (Kip et al., 2010; Liebner et al., 2011; Raghoebarsing et
al., 2005). Studies by Nichols et al. (2009) show that a higher water level at the peat surface is crucial
for   high methane-derived $CO_2$ release. Furthermore, symbiosis with methanotrophs enhances the
moisture-related effect on the $\delta^{13}C$ signal of bryophytes. Endophytic microorganisms in hyalocytes of
submerged *Sphagnum* (Raghoebarsing et al., 2005) or epiphytic microorganisms on submerged brown
mosses (Liebner et al., 2011) are presumed to provide $^{13}C$-depleted $CO_2$ directly to the lamina cells of
mosses. The studies of Ruttner (1947) and Bain and Proctor (1980) show that, in general, moss taxa
are incapable of bicarbonate uptake. Hence, bicarbonate, known to be a carbon source for submerged
vascular plants and algae (Herzschuh et al., 2010; Merz, 1992), can most probably be excluded as a
carbon source for moss and thus as a factor influencing the $\delta^{13}C$ value. Moreover, the bicarbonate
content in pond waters in northern Siberian landscapes is very low (Wetterich et al., 2008). Other
sources of $^{13}C$-depleted $CO_2$ are surface run-off during spring flooding, rain events and decomposition
processes in the pond (Leng and Marshall, 2004; Maberly et al., 2013). Yet we cannot fully eliminate
the possibility that the measured bulk material was contaminated in parts with epiphytic or endophytic
microorganisms. The overall isotopic composition would, however, likely be unaffected, as Ménot and
Burns (2001) have shown that the $\delta^{13}C$ values of bulk organic material and alpha-cellulose of
*Sphagnum* are very similar. Thus, the large ranges within several species of meso-hgryophilic habitats
in arctic regions suggest that the existence of open water leads to more depleted $\delta^{13}C$ values and
measurements of the isotopic composition of methane when present and microbial groups in the water
and terrestrial litter should be possible. Finally, considering the relationship of selected brown mosses
to mean plant position, the complex origin of plant-available carbon makes it difficult to interpret the
$\delta^{13}C$ record, especially for meso-hygrophilic brown mosses as well as *Sphagnum* (Prince et al., 1997,
Ménot and Burns, 2001).

## 4.3 $\delta^{15}N$ isotopes

Like $\delta^{13}C$, the interpretation of stable nitrogen isotope compositions of mosses is challenging because,
again, source signals need to be separated from those originating from physiological isotopic
discrimination processes. Our results yield relatively $^{15}N$-depleted $\delta^{15}N$ values for xero-mesophilic
mosses growing preferentially on rims compared to meso-hygrophilic mosses.
The terrestrial arctic systems are generally thought to be nitrogen limited (Gordon et al., 2001;
Kielland, 1997). On the rim sites, atmospheric deposition can be considered to be the most important
source for nitrogen (Jonasson and Shaver, 1999) originating from fog, dew, precipitation and surface
run-off (Sveinbjörnsson and Oechel, 1992). However, most of the nitrogen available to rim mosses
originates from recycling of already $^{15}N$-depleted higher plant and moss litter (Turetsky, 2003). The
ectohydric morphology enables an efficient nutrient uptake across the entire moss plant surface via
trapped water. In fact, the meso-hygrophilic group has a higher N content than the xero-mesophilic
group (see section on C/N ratio). Inorganic nitrogen, but especially the high amounts of organic
nitrogen provided by N-mineralization in tundra soils (Kielland, 1995) are important for mosses
growing on the rather dry sites such as the polygonal rims (Atkin, 1996).
Three of the investigated submerged or floating moss species show a significant positive
relationship between water-level and $\delta^{15}N$ values. These results are similar to those of Asada et al.
(2005) who tested a relationship between $\delta^{15}N$ values of different *Sphagnum* species and their position

relative to the groundwater level, which they assumed to originate from different nitrogen sources and different internal fractionating processes. We assume that the often heavier nitrogen isotope composition of meso-hygrophilic brown moss individuals originates from the high degree of symbiotic associations with aquatic atmospheric nitrogen-fixing autotrophic microorganisms such as *Nostoc* or *Anabena* (Lindo et al., 2013) or methanotrophs (Vile et al., 2014). The high spatial degree with endo- or ectosymbiotic $N_2$-fixing microorganisms enables the direct uptake of their nitrogen products, which is similar to that of $N_2$ in air.

## 4.4 *n*-alkane patterns

Compared to vascular plants that are characterized by a thick leaf-wax layer, mosses produce only a small amount of *n*-alkanes (Baas et al., 2000; Ficken et al., 1998). Like previous studies on vascular plants (Aichner et al., 2010; Ficken et al., 2000; Meyers and Ishiwatari, 1993), our results generally reveal a differentiation between terrestrial taxa (i.e. xero-mesophilic group) characterized mainly by *n*-alkanes maximizing at $n\text{-}C_{29}$ and $n\text{-}C_{31}$ and submerged living taxa (i.e. the meso-hygrophilic group) maximizing at $n\text{-}C_{25}$ and $n\text{-}C_{27}$. Earlier investigations of Nott et al. (2000), Baas et al. (2000) and Bingham et al. (2010), who compared the *n*-alkane fingerprints of *Sphagnum* taxa growing along a hydrological gradient, agree with our results.

Huang et al. (2012a) and Ficken et al. (1998) used proxy ratios (ACL, $P_{aq}$) to divide moss taxa roughly by their moisture preferences. They calculate the $ACL_{23\text{-}33}$ for samples of lichens and *Racomitrum lanuginosum*, which have rather similar hydrological requirements to our xero-mesophilic mosses. Despite the slightly narrower $ACL_{23\text{-}33}$ range, their results show similarities to our xero-mesophilic group. A comparison between $ACL_{21\text{-}33,}$ ratios of *Sphagnum* (plant position nearly at water-level) of Huang et al. (2012b) and our brown mosses shows that the $ACL_{21\text{-}33}$ ratios of *Sphagnum* species are rather lower. The intraspecific conclusion of Huang et al. (2012b) (wetter moisture conditions entail lower $ACL_{21\text{-}33}$) for *Sphagnum* is reflected by our measurements.

**dry** ($ACL_{21\text{-}33} = 29.1 - 27.5$)     **< moisture condition >**     ($ACL_{21\text{-}33} = 26.4 - 25.2$) **wet**
   Tom_nit   <   Aul_tur   <   Aul_pal   <   Dre_rev   <   Sco_sco

.

As we observed a clear difference in the $ACL_{21-33}$ between the xero-mesophilic and the meso-
hygrophilic group, we suggest that the inclusion of mid-chain $n$-alkanes ($n$-$C_{21}$ to $n$-$C_{25}$) in the
equation of ACL improves its value as a proxy for moisture conditions. Andersson et al. (2011)
inferred $ACL_{27-31}$ values of 29 for brown-moss peat from western Russian during wet phases, which is
however, poorly comparable to our results because they investigated total peat organic matter instead
of pure moss material.
Ficken et al. (2000) proposed $P_{aq}$ as a semi-quantitative proxy ratio for the differentiation of
terrestrial and aquatic plants (<0.1 terrestrial plants, 0.1–0.4 emergent macrophytes, 0.4-1
submerged/floating macrophytes). Our inferred $P_{aq}$ results for the individual species agree with these
assumptions. If we consider that the proxy ratio levels were created by vascular plants from a limited
dataset of lakes in Kenya and as we focus on non-vascular plants of the arctic, we chose other level
terms.

| terms by Ficken et al. (2000) | emergent macrophytes | | submerged/floating macrophytes | | |
|---|---|---|---|---|---|
| adapted terms for mosses | xero-mesophilic mosses | | meso-hygrophilic mosses | | |
| species sorted by $P_{aq}$ | Aul_tur  <  Tom_nit | < | Aul_pal  <  Dre_rev | < | Sco_sco |


Overall, our results do not support the inference of Nichols et al. (2006) that a hydrological
classification is possible between *Sphagnum* and non-*Sphagnum* formed peat as the latter show wide
variations between different habitats. The inferred broad $P_{aq}$ range of *Drepanocladus* and between
both *Aulacomnium* probably indicates that intraspecific variation is related to the individual´s growing
condition, which could provide the basis to develop $P_{aq}$ as a proxy for water-level when measured on
taxonomically identified fossil plant material.
As with $P_{aq}$, $n$-alkanes seem to be species-specific given stable environmental parameters and are
related to the species-specific moisture requirements, which are adapted to changing environmental
conditions. Thus, our results confirm the conclusions of Bingham et al. (2010), Bush and McInerney
(2015) and Nott et al. (2000) that the pattern of $n$-alkanes has the potential to become a valuable proxy
for chemotaxonomic identification and moisture conditions. *Scorpidium scorpioides,* a species with a
rather narrow preference range (i.e. it is limited to open water conditions), shows low intraspecific
variations. This matches the results for *Sphagnum* compiled by Bingham et al. (2010), which also
show minor intraspecific variations. *Aulacomnium* in contrast, which grows in a rather wide range of
moisture conditions, shows strong variations in its *n*-alkane spectra: whether this is a function of the
individual's growing conditions, however, needs to be investigated in a more extensive study.

## 5. Conclusions

The habitat and intraspecific isotopic and chemical patterns of 10 brown-moss species detected along
small-scale hydrological gradients in Siberian polygonal tundra were studied.

393       The observed higher $C/N_{(m)}$ ratios of xero-mesophilic mosses compared to those of the meso-

hygrophilic mosses originate from the different environmental requirements when living emergent (i.e.
investment in a higher stability resulting in high C/N ratios) as opposed to submerged. Furthermore,
the latter group may also gain a better nitrogen supply through microbial symbioses.

397       With respect to the isotopic source pools, the meso-hygrophilic species have greater access than

xero-mesophilic species, which is seen in their large ranges. The approximate habitat-specific division
of $\delta^{13}C$ values as a result of discrimination by RuBisCO under different hydrological regimes is
overturned by the influence of different sources and cannot provide a clear distinction from a single
measurement of either habitat type. For species, growing near the water level, no intraspecific
relationship with water level was observed probably as a result of the parallel impact of processes
causing opposing $\delta^{13}C$ trends.

404       Our analyses reveal that, compared with xero-mesophilic mosses, meso-hygrophilic mosses are

characterized by enriched $\delta^{15}N$ values probably originating from microbial symbioses. Both carbon
and nitrogen isotopic ratios seem to be valuable proxies to differentiate between taxa preferring the
polygon rim or pond. Moreover, with respect to meso-hygrophilic mosses, the detected positive
relations between intraspecific variations and the individuals' relative growing position could allow
even more semi-quantitative information about water-level changes to be inferred.
The *n*-alkane patterns of brown mosses (limited 16 individuals belonging to five species) indicate
that they are species-specific and have thus the potential to be developed as a chemotaxonomic proxy.
The applicability of proxy ratios (ACL and Paq) could be attested for arctic mosses after adjustments
of the levels.
Overall, our study indicates that C/N, isotopic and *n*-alkane analyses of brown moss material has
a high environmental indicator potential, particularly if species-specific material instead of bulk
material is analysed.

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

## Figures

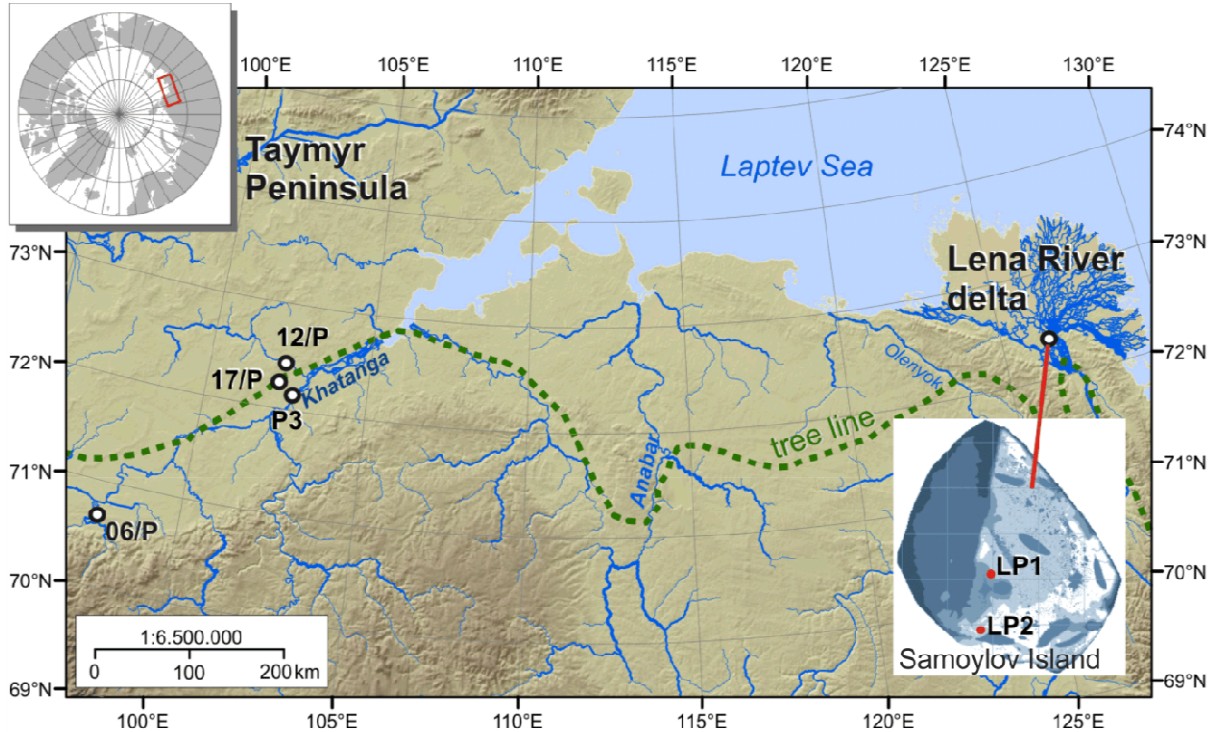

Fig. 1: The Khatanga study sites are located in the south-east of the Taymyr Peninsula and both the sites on Samoylov Island are in the southern Lena River delta (Map by Th. Böhmer).

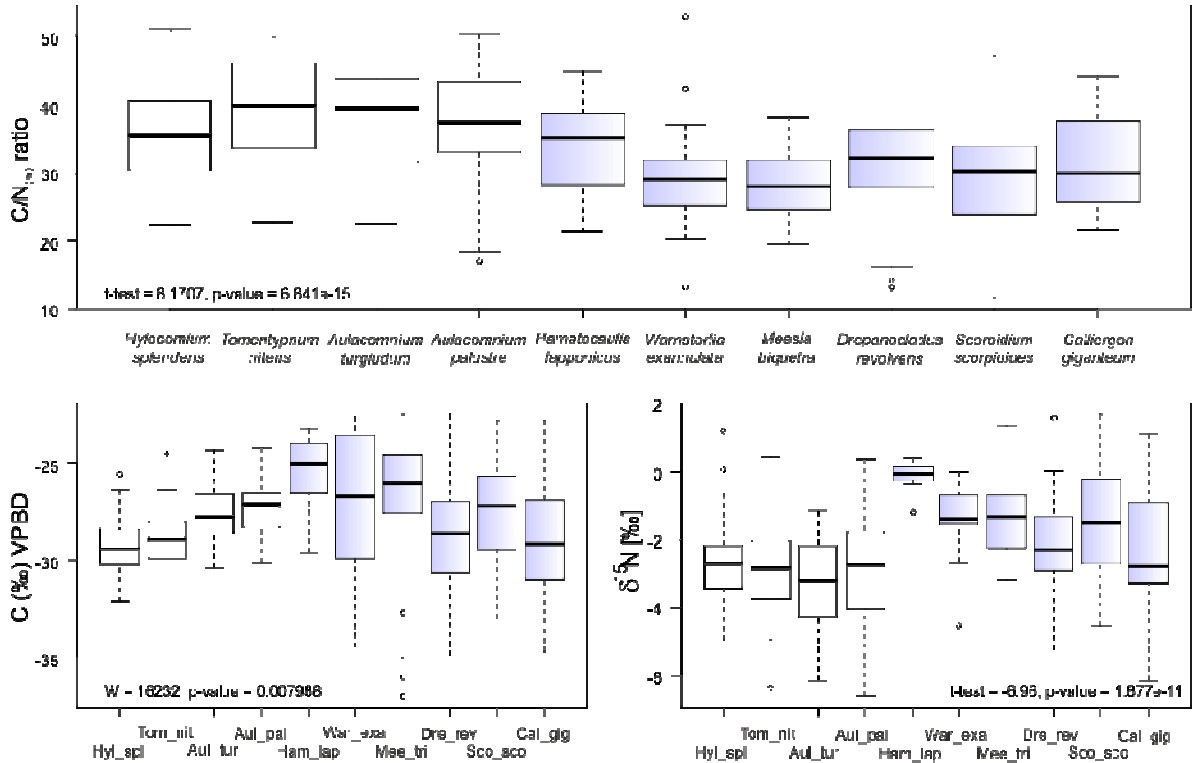


**Fig. 2:** *C/N (m) ratios, $\delta^{13}C$ and $\delta^{15}N$ values of the 10 moss taxa studied. White boxplots are the xero-mesophilic group and boxplots shaded in blue are the meso-hygrophilic group. A t-test was done to distinguish the signals between the two habitat groups.*


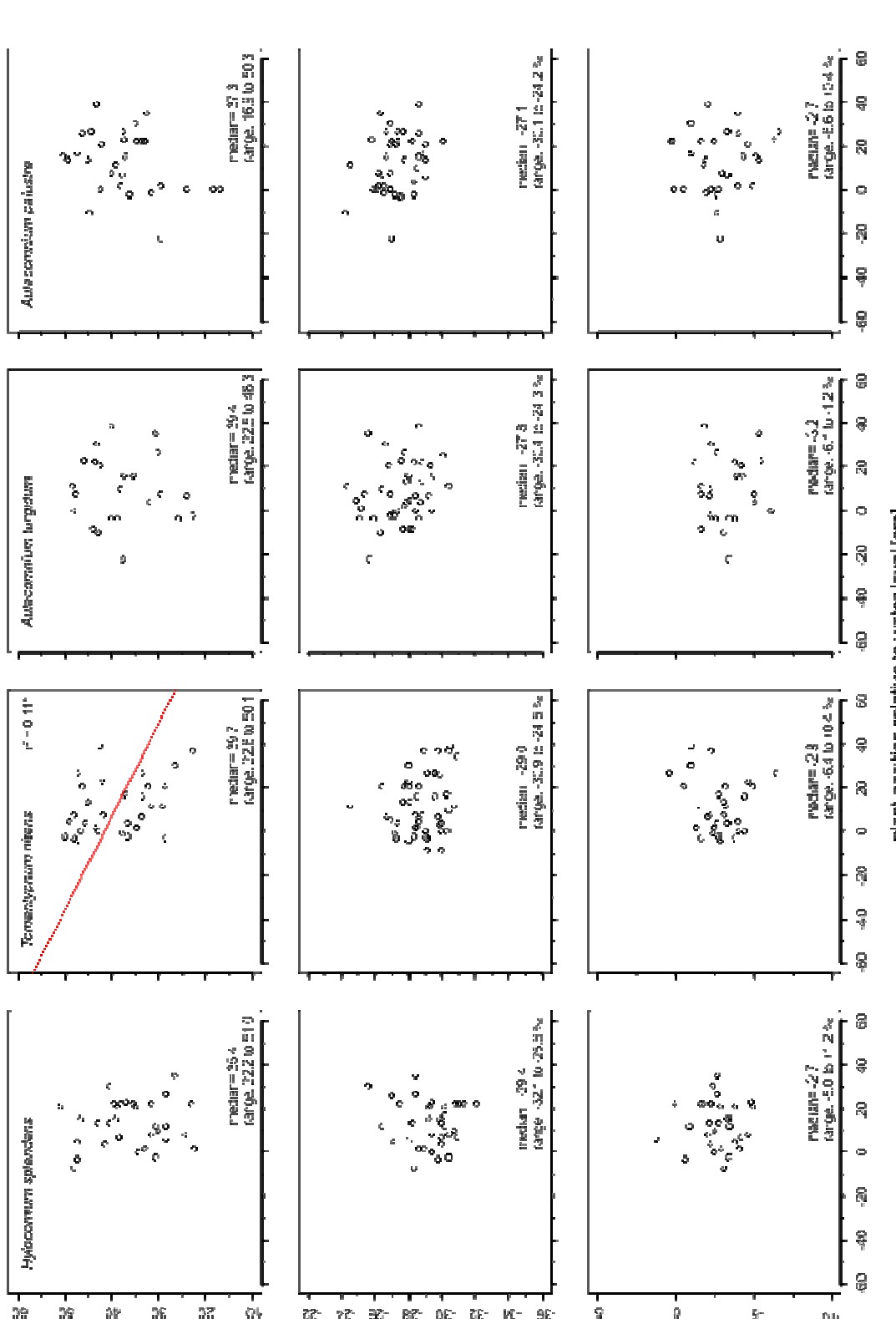

**Fig. 3a:** *Intraspecific relationships of the C/N$_{(m)}$ ratio and the stable isotope values of carbon and nitrogen related to the water-level of the xero-mesophilic moss group. Regression lines (red) are only plotted for significant data sets.*


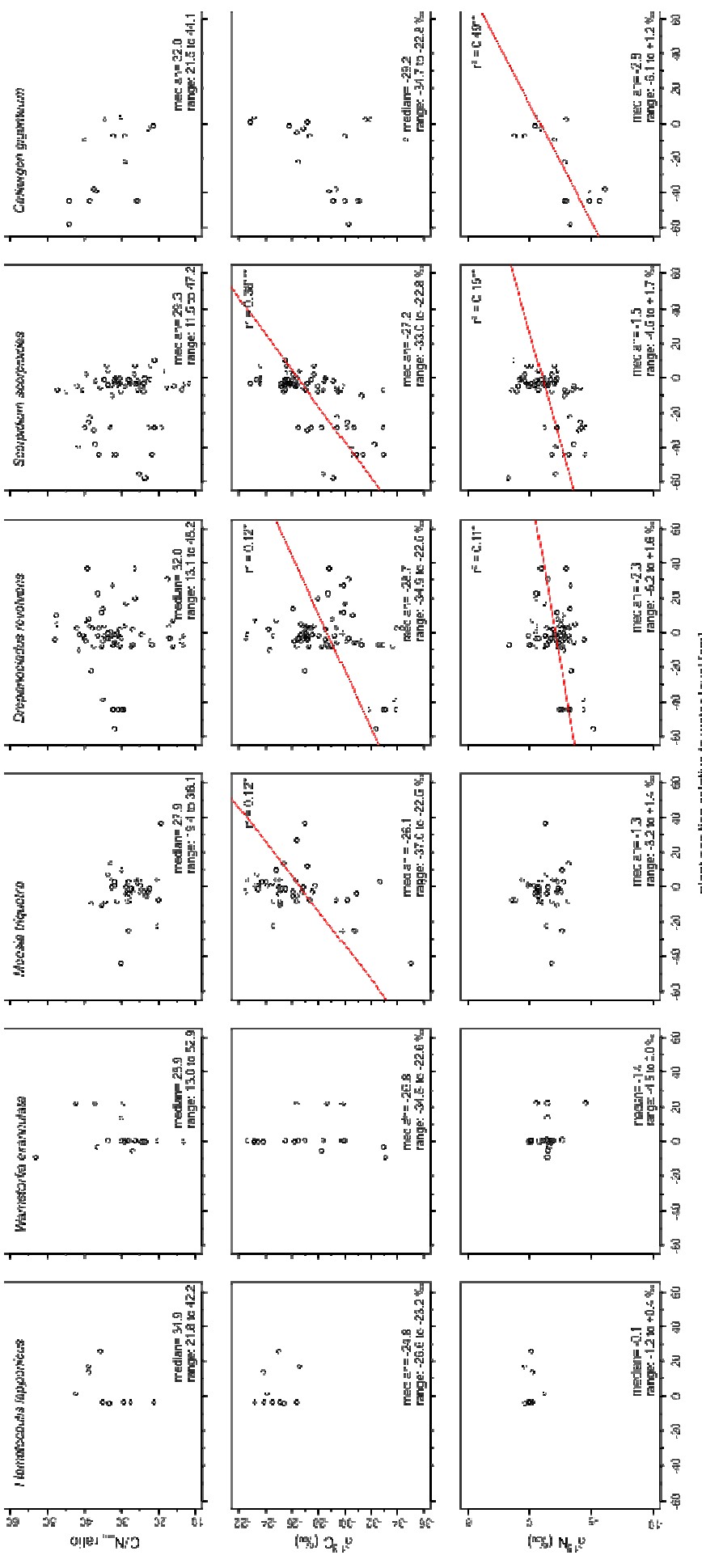

**Fig 3b:** *Intraspecific relationships of the C/N$_{(m)}$ ratio and the stable isotope values of carbon and nitrogen related to the water-level of the meso-hygrophilic moss group.*
*Regression lines (red)are only plotted for significant data sets.*

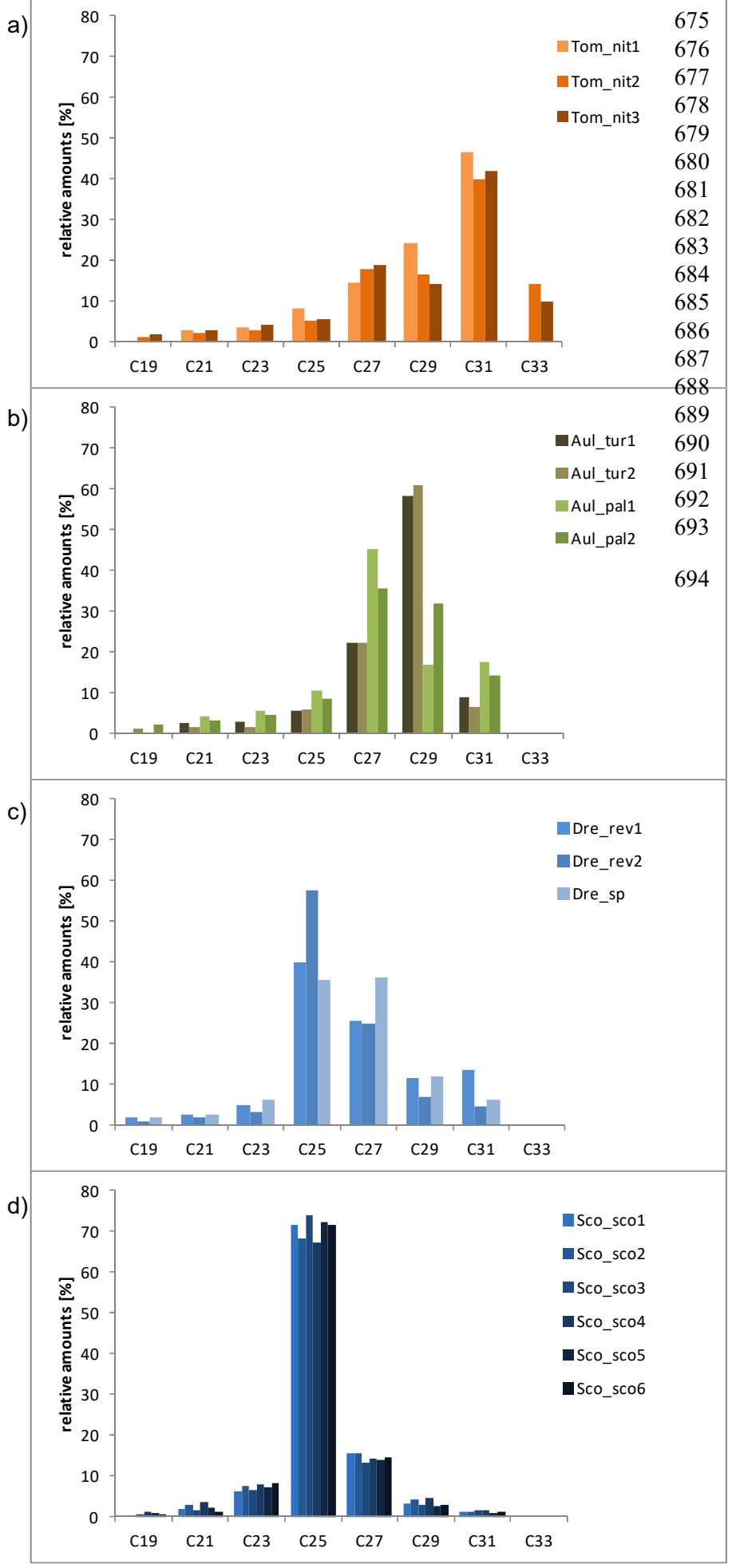



*Fig. 4:* *The relative amounts [%] of* n-*alkanes with an odd number of carbon atoms in selected brown mosses grouped by species and ordered by their preferences with respect to their mean plant-position relative to water-level for xero-mesophilic (a, b) and meso-hygrophilic (c, d) mosses. Tom-nit: Tomentypnum nitens; Aul_tur: Aulacomnium turgidum; Aul_pal: Aulacomnium palustre; Dre_rev: Drepanocladus revolvens; Dre_sp: Drepanocladus sp.; Sco_sco: Scorpidium scorpioides*

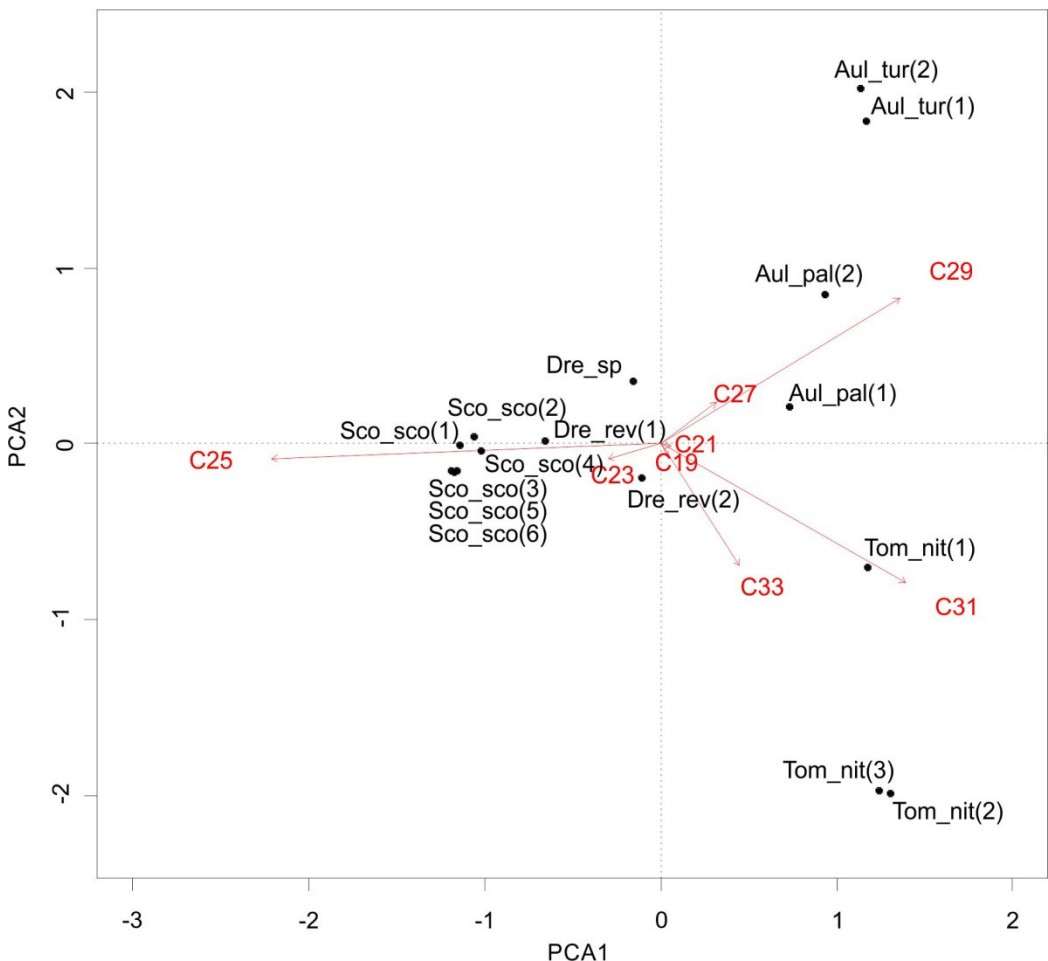


**Fig 5:** *PCA of* n-*alkanes separates the meso-hygrophilic group on the left side of axis 1 from the xero-mesophilic group on the right side. Along the second axis the* Aulacomniaceae *are distinguished from* Tomentypnum nitens *according to the distribution of long-chain* n-*alkanes with more or less than 30 carbon atoms.*


# Appendix

***Table A1:*** *Brief description of studied polygons. (For further information see Zibulski et al., 2016)*

| polygon cross section, (length of transect = polygon size) | | coordinates | short description vegetation type; additional information |
|---|---|---|---|
|  | 06/P | 70.666° N, 97.708° E | open forest; swinging bog (mat of mosses) |
|  | 17/P | 72.243° N, 102.233° E | forest-tundra intersection; shallow and sparsely vegetated |
|  | P3/I | | forest-tundra intersection; a complex of three individual polygons: P3/I - deep, open water body P3/II - shallow, open water body P3/III - shallow completely vegetated |
|  | P3/II | 72.149° N, 102.693° E | |
|  | P3/III | | |
|  | 12/P | 72.431° N, 102.373° E | tundra; shallow and vegetated |
|  | LP1 | 72.375° N, 126.483° E | tundra; deep polygon without thaw depth below the water body |
|  | LP2 | 72.370° N, 126.481° E | tundra; shallow and vegetated |


***Table A2:*** *C/N(m) ratio, $\delta^{13}C$ and $\delta^{15}N$ data of individual mosses depending on their position relative to the water-level (cm).*

| Species | C/N(m) ratio | | $\delta^{13}C$ [‰] | | | $\delta^{15}N$ [‰] | | |
| --- | --- | --- | --- | --- | --- | --- | --- | --- |
| | median | range | n | median | range | n | median | range |
| *H. splendens* (Hyl_spl) | 47.1 | 29.5 to 67.9 | 34 | -29.4 | -32.1 to -25.6 | 31 | -2.7 | -5.0 to +1.2 |
| *T. nitens* (Tom_nit) | 52.9 | 30.1 to 66.6* | 46 | -29.0 | -30.9 to -24.5 | 30 | -2.8 | -6.4 to +0.4 |
| *A. turgidum* (Aul_tur) | 52.4 | 29.9 to 64.3 | 41 | -27.8 | -30.4 to 24.3 | 24 | -3.2 | -6.1 to -1.2 |
| *A. palustre* (Aul_pal) | 49.6 | 22.5 to 66.9 | 40 | -27.1 | -30.1 to -24.2 | 30 | -2.7 | -6.6 to +0.4 |
| *H. lapponicus* (Ham_lap) | 46.5 | 28.3 to 56.1 | 10 | -24.8 | -26.6 to -23.2 | 9 | -0.1 | -1.2 to +0.4 |
| *W. exannulata* (War_exa) | 38.4 | 17.3 to 70.4 | 20 | -26.8 | -34.5 to -22.6 | 19 | -1.4 | -4.5 to 0.0 |
| *M. triquetra* (Mee_tri) | 37.1 | 25.8 to 50.7 | 45 | -26.1 | -37.0 to -22.5* | 34 | -1.3 | -3.2 to +1.4 |
| *D. revolvens* (Dre_rev) | 42.6 | 17.5 to 64.1 | 72 | -28.7 | -34.9 to -22.5* | 67 | -2.3 | -5.2 to +1.6** |
| *S. scorpioides* (Sco_sco) | 38.9 | 15.4 to 62.8 | 69 | -27.2 | -33.0 to -22.8*** | 65 | -1.5 | -4.5 to +1.7** |
| *C. giganteum* (Cal_gig) | 42.6 | 28.6 to 58.7 | 23 | -29.2 | -34.7 to -22.8 | 17 | -2.9 | -6.1 to +1.2** |

Stars designate significant linear regressions between parameter and the plant position relative to water-level (* p≤0.01, ** p≤0.05,
*** p≤0.001).