# Peer review of "C/N ratio, stable isotope (δ13C, δ15N), and *n*-alkane patterns of brown mosses along hydrological gradients of low-centred polygons of the Siberian Arctic Romy Zibulski1,2, Felix Wesener4, Heinz Wilkes3,5, Birgit Plessen3, Luidmila A. Pestryakova6, Ulrike Herzschuh1,2,7"

_Biogeosciences, 2016_

## Referee Comment (RC1) · Anonymous Referee #1 · 11 Sep 2016

Title. why not use brown mosses or equivalent instead of bryophytes?

Number of samples: In line 20 you speak about 400 samples from 10 species, and in line 22 'six of these taxa', it not clear how many samples (from those 400) were investigated for n-alkanes.

Table A2: why is there so much variation in the number of samples analysed (n)? n for 15N is lacking

Line 35 is '... as a function of microbial symbiosis' also an assumption? If not from which data is this concluded?

The Title and Abstract suggest that you investigate the effects of a hydrological gradient on 13C, 15N, C/N and n-alkane distribution, but from the paper it remains unclear to me if this really has been done, i.e., it seems that certain species are labelled meso-hygrophilic/xero-mesophylic, and compared as such, but the same species may occur in different habitats (water levels) and compared according to habitat (as is suggested by the title). I would recommend to make this explicitly clear, already in the Abstract but also throughout the ms. So, make clear how you define meso-hygrophilic mosses, are these well-known dry habitat mosses OR are you looking at the difference between individuals in a certain habitat?

Lines 37-39. Be more specific. 'Isotopic and biochemical signals' are mentioned, I recommend using C/N and n-alkanes instead of biochemical. Also 'certain moss taxa' could you mention them?

Lines 37-39. I would be more careful in exptrapolating the results to be useful in paleaoenvironmental studies, this is not the subject of the ms, and the results do not enable such discussion because peat or SOM is mostly decomposed and a mixture of several species, it is therefore not valid to state that the effect of hydrological conditions on the here analysed proxies is dominating that of species, or decomposition. Also line 52-55, I think that the ms should not focus on paleoenvironemtnal interpretation, as no such data are provided and it is thus only distracting from the subject; I recommend not to focus on this, but mention in the Conclusion that this may be a problem for the interpretation of these proxies in paleostudies and the need for future research on this.

Line 37 '…..) and intermediate (C25) chain length, respectively.'

Line 49. Do you mean mosses in general with 'their'? the abundance of mosses in peat and permafrost is probably more related to ecology (cold wet conditions), instead of to its intrinsic low decomposition rate.

Lines 51-52. I don't think that little is known about stable isotopes and 'biochemical characteristics' of mosses, the effect of this study (habitat) is very interesting and indeed not much of it is known.In the next sentence (line 56-58) you say that these are the most commonly used parameters, isnt'that contradicting?

I recommend to use 'n-alkane and C/N ratio' instead of 'biochemical characteristics' throughout the paper. Because the term biochemical characteristics is much broader.

Line 60 microbial material instead of activity?

Line 75. Explicitly mention this point earlier (already in the Abstract), because mosses and peatlands are often associated with Sphagnum. Indeed this study is more novel especially of the focus on non-sphagnum mosses, but this is not clear from the beginning and may also solve the other problem mentioned above.

Line 104. This is a very good idea, I would mention this already in the Abstract, see earlier comments. But then I see the Methodology in lines 124-131 and the question arises if this really has been studied because mean values are used to determine its hydrological habitat, thereby loosing important information on the effect of water level. Relevant information is missing on how many plants were studied for each of these means and how is the variation within each group?? It is again not clear to me how the the two groups (xero-meso and meso-hygro) are defined, on the basis of species or hydrological habitat?

Line 135 abbreviations have already been used

Line 178 perhaps individual instead of single?

Line 188 Fig. 3b?

Line 190. If the alkanes are meant to use as a proxy for species or water level, then the data of absolute abundance must be presented as well, as this is highly variable like you indicate here. In a mixture of species like in peat and soil, a certain distribution can only be related to the species (or water level) if differences in absolute abundance is taken into account.

Line 195 I don't understand this. According to the figure there is not really a species-preferred position all a-d have a -3 and 10 for example. If you look at the highly variable water levels within each species in Fig. 4, then I would not say that the species can be separated into those groups of xero and mesophilic. It must be more explicitly mentioned how these groups are defined, based on what.

I would delete the enter between lines 196 and 197

Lines 225-226. Now I understand it better. This is a pity. It would be very interesting do such a study, correlate the water level to the plant chemistry, at the plant detail, not the plot. It should be made very clear in the Method Section, now I had the impression that it was done on the individual plant level.

Line 228-229 this must be mentioned must earlier.

Line 297 but there is a large variability. See line 195

Line 329, I do not understand the word choice 'individuals' in the context of the sampling design, see earlier comments

Line 333 no fossil material was studied, so this conclusion cannot be made

―――――――――――――――――

---

## Referee Comment (RC2) · P.A. Meyers (Referee) · 19 Sep 2016

Zibulski and colleagues present the results of their survey of organic geochemical properties of brown mosses collected over hydrologic gradients in the Siberian Arctic. Their survey is impressively extensive, consisting of 400 samples that includes ten species. The parameters that they have measured is also impressive and comprise C/N ratios, stable carbon and nitrogen isotope values, and n-alkane distributions. To render some sense out the mass of data that they have generated, the authors performed a principal component analysis. Patterns emerge from the data. Plants from wetter settings have

somewhat different sets of properties than plants from less wet settings, but intraspecific variability obscures simple conclusions. Although the study was conducted at a somewhat exotic location, it has significance to paleoenvironmental and paleoclimatic studies at many other locations that have important contributions from mosses and is appropriate for publication in Biogeosciences. However, several technical details need attention, and the data in general need more interpretation.

The foremost issue is that the n-alkane data are inadequately presented and interpreted. As correctly noted by the authors, mosses typically have lower concentrations of these wax components than vascular land plants. Nonetheless, the absolute concentrations of all 400 samples should be reported in the supplemental material and summarized in either a figure or a table in the manuscript proper. Furthermore, the relative concentrations of the samples should be compared using some of the well established n-alkane ratios such as the Paq, the ACL, and the CPI. For the ACL., I recommend using an extended range (21-33), similar to the extended range used by Bush and McInerney (2015, Org Geochem 79, 65-73). Addition of these ratios will allow better comparison of these new results to results from existing and future studies of wetland biogeochemical records, and it will likely enable the authors to refine their interpretations.

Another issue is that the authors do not make full use of their important documentation of the intraspecific variability in the geochemical properties of these plants. Other studies have found similar variability in both Sphagnum (Huang et al., 2012, Org Geochem 44, 1-7) and in vascular plants (Feakins et al, 2016, Org Geochem 100, 89-199), and they both discuss its possible significance and mention other reports of intraspecific variability. Better appreciation of this variability is important to better interpretation of the paleoenvironmental significance of these properties, and the authors' data could make a better contribution to such understanding that it presently does.

Yet another issue is that aspects of this study seem to provide answers, at least partial, to some of the questions raised by Andersson et al. (2011, Org Geochem 42, 1065-

1075; 2012, J. Quat. Sci. 27, 545-552) from their study of the changes in C/N ratios, stable carbon and nitrogen isotope values, and n-alkane distributions recorded in the fen-bog transition of a peat sequence in the Russian Arctic. Consideration of these questions and how the new data may address them would enrich both the Introduction and the Discussion of this contribution.

Finally, the authors need to specify in the text and figures whether the C/N ratios that they report are atomic or weight ratios.

Phil Meyers, September 19, 2016

---

## Author Comment (AC1) · 11 Nov 2016

*Reply to reviewers' comments concerning the manuscript:*

*"C/N ratio, stable isotope ($\delta^{13}C$, $\delta^{15}N$), and n-alkane patterns of bryophytes along hydrological gradients of low-centred polygons of the Siberian Arctic"* by R. Zibulski et al.

**Referee #2**

*Received 19th September 2016*

We thank reviewer #2 for the valuable comments.

**comments:**

*The foremost issue is that the n-alkane data are inadequately presented and interpreted. As correctly noted by the authors, mosses typically have lower concentrations of these wax components than vascular land plants. Nonetheless, the absolute concentrations of all 400 samples should be reported in the supplemental material and summarized in either a figure or a table in the manuscript proper. Furthermore, the relative concentrations of the samples should be compared using some of the well established n-alkane ratios such as the Paq, the ACL, and the CPI. For the ACL., I recommend using an extended range (21-33), similar to the extended range used by Bush and McInerney (2015, Org Geochem 79, 65-73). Addition of these ratios will allow better comparison of these new results to results from existing and future studies of wetland biogeochemical records, and it will likely enable the authors to refine their interpretations.*

Our response: In the original version of the manuscript we indicated that only 16 samples were selected for *n*-alkanes analyses because these samples were also used for other analyses. Thus, we consider this a preliminary study. We now include Table 2 with the absolute concentrations of all 16 *n*-alkanes samples and added the $P_{aq}$ of Ficken et al. (2000) and the ACL with the recommended range *n*-$C_{21}$ to *n*-$C_{33}$ ($ACL_{21-33}$). During the carbon chain analyses of the 16 samples, there were selected 10 further samples for advanced analysis of even-chain homologues. Thus, we do not calculate the CPI.

This entailed changes in the Methods section for *n*-alkanes (added the formula), in the Results section (reference to Table 2), and in the Discussion section '4.4 n-alkane patterns'.

Methods*: Additionally, we calculated the average chain length ($APL_{21-33}$), assumed to represent a proxy for moisture (Andersson et al, 2011) and temperature (Bush and McInerney, 2015), with a comprehensible extended range from n-$C_{21}$ to n-$C_{33}$ and the proxy ratio $P_{aq}$, which was developed as a proxy ratio to distinguish submerged or floating aquatic macrophytes from emergent and terrestrial plants (Ficken et al., 2000):*

$$APL_n = \frac{\sum(n \times C_n)}{\sum C_n} \quad , n = 21 - 33$$

$$P_{aq} = \frac{(C_{23} + C_{25})}{(C_{23} + C_{25} + C_{29} + C_{31})}$$

Results: *Evaluations of the n-alkane biomarker proxies, $ACL_{21-33}$ and $P_{aq}$, also show a clear division*
*between the xero-mesophilic and the meso-hygrophilic species groups (Table 2), whereas intraspecific*
*variations are low (with the exception of Drepanocladus). The xero-mesophilic group is notable for*
*high averages of the $APL_{21-33}$ (28.41) and $P_{aq}$ (-0.06) compared to low averages of $APL_{21-33}$ (25.61)*
*and $P_{aq}$ (-0.7) for the meso-hygrophilic group.*

Discussion: We added citations and reworded the Discussion section completely.

***Another issue is that the authors do not make full use of their important documentation of the***
***intraspecific variability in the geochemical properties of these plants. Other studies have found***
***similar variability in both Sphagnum (Huang et al., 2012, Org Geochem 44, 1-7) and in vascular***
***plants (Feakins et al, 2016, Org Geochem 100, 89-199), and they both discuss its possible***
***significance and mention other reports of intraspecific variability. Better appreciation of this***
***variability is important to better interpretation of the paleoenvironmental significance of these***
***properties, and the authors' data could make a better contribution to such understanding that it***
***presently does.***

Our response: We cite Huang et al. 2012 in the discussion of *n*-alkanes. The study of Feakins et al.
(2016) concerns an analysis along a temperature gradient. This is not so relevant to our study as our 16
samples of n-alkanes come from one site, as explained in the Methods section 2.4.

***Yet another issue is that aspects of this study seem to provide answers, at least partial, to some of the***
***questions raised by Andersson et al. (2011, Org Geochem 42, 1065-C21075; 2012, J. Quat. Sci. 27,***
***545-552) from their study of the changes in C/N ratios, stable carbon and nitrogen isotope values,***
***and n-alkane distributions recorded in the fen-bog transition of a peat sequence in the Russian***
***Arctic. Consideration of these questions and how the new data may address them would enrich both***
***the Introduction and the Discussion of this contribution.***

Our response: We think this suggestion is justified with regard to meaningful palaeo-reconstructions in
fens with a dominance of brown mosses. Both articles show interesting studies in a changing wetland
with a dominant portion of brown mosses in the lower fen part. However, due to the different
approach of Andersson et al. and especially the different experimental setup with measurements of
bulk instead of species-specific material (e.g. only minor quantities of vascular plant material mask the
signal of *n*-alkanes$_{moss}$ with their higher amount of *n*-alkanes, and stable isotope properties are changed
by physical fractionation within a soil column) cause problems for a meaningful comparison and
discussion of contrary results. Furthermore, we agree with reviewer #1, who noticed that only recent
material is measured and thus that interpretations and suggestions about fossil material could only be
speculative. However, we added some information and comparisons.

Addition to Introduction: *We provide C/N ratios by weight of arctic brown mosses, in the anticipation*
*that they will be useful for comparative palaeo-environmental reconstructions (Andersson et al. 2011,*
*) and in the evaluation of organic matter sources in Russian permafrost soils with regard to species*
*and habitat-specific patterns.*

*Addition to Discussion: Andersson et al. (2011) inferred $ACL_{27-31}$ values of 29 for brown-moss peat*
*from western Russian during wet phases, which is however, poorly comparable to our results because*
*they investigated total peat organic instead of pure moss material.*

**Finally, the authors need to specify in the text and figures whether the C/N ratios that they report**
**are atomic or weight ratios.**

Our response: We thank you for this comment and have indicated the '*weight'* information throughout
$(C/N_{(wt)})$.

[revised manuscript text omitted]
_{21-33}$) for *Sphagnum* is reflected by our measurements for *Drepanocladus* and *Scorpidium*, i.e. they show a dependence with water level. As we observed a clear difference in the $ACL_{21-33}$ between the xero-mesophilic and the meso-hygrophilic group, we suggest that the inclusion of mid-chain $n$-alkanes ($n$-$C_{21}$ to $n$-$C_{25}$) in the equation of ACL improves its value as a proxy for moisture conditions. Andersson et al. (2011) inferred $ACL_{27-31}$ values of 29 for brown-moss peat from western Russian during wet phases, which is however, poorly comparable to our results because they investigated total peat organic matter instead of pure moss material.

Ficken et al. (2000) proposed $P_{aq}$ as a semi-quantitative proxy ratio for the differentiation of terrestrial and aquatic plants (<0.1 terrestrial plants, 0.1–0.4 emergent macrophytes, 0.4-1 submerged/floating macrophytes). Our inferred $P_{aq}$ results for the individual species fit well with these assumptions. For example, submerged/floating *Scorpidium* ($P_{aq}$ median: 0.56) has a higher median $P_{aq}$ value than floating/mesic *Drepanocladus* ($P_{aq}$ median: 0.34), which is again higher than that of the xero-mesophilic mosses *Tomentypnum* ($P_{aq}$ median: 0.13) and *Aulacomnium* ($P_{aq}$ median: 0.13). Ficken et al. (2000) also measured a *Drepanocladus* sample ($P_{aq}$ = 0.30). 
[revised manuscript text omitted]

---

## Author Comment (AC2) · 11 Nov 2016

*Reply to reviewers' comments concerning the manuscript:*

"***C/N ratio, stable isotope ($\delta^{13}C$, $\delta^{15}N$), and n-alkane patterns of bryophytes***

***along hydrological gradients of low-centred polygons of the Siberian Arctic***"

by R. Zibulski et al.

***Anonymous Referee #1***

*Received 11$^{th}$ September 2016*

We thank reviewer #1 for the valuable comments which have contributed to the improvement of the article.

**general comments:**

***Title. why not use brown mosses or equivalent instead of bryophytes?***

Our response: We changed the title as follows:

"C/N ratio, stable isotope ($\delta^{13}C$, $\delta^{15}N$), and n-alkane patterns of *brown mosses along hydrological*

*gradients of low-centred polygons of the Siberian Arctic*"

***The Title and Abstract suggest that you investigate the effects of a hydrological gradient on 13C,***

***15N, C/N and n-alkane distribution, but from the paper it remains unclear to me if this really has***

***been done, i.e., it seems that certain species are labelled meso-hygrophilic/xero-mesophylic, and***

***compared as such, but the same species may occur in different habitats (water levels) and compared***

***according to habitat (as is suggested by the title). I would recommend to make this explicitly clear,***

***already in the Abstract but also throughout the ms. So, make clear how you define meso-hygrophilic***

***mosses, are these well-known dry habitat mosses OR are you looking at the difference between***

***individuals in a certain habitat?***

Our response: We accept the reviewer's comment and have refined the Methods section 'Sampling and studied moss species'. The parameter h is not the absolute plant position in relation to water-level, it is the mean plant position calculated from the plot position in relation to water-level of all individuals for each studied species as an approximation. The classification of habitat types is based on our findings during the field expedition.

**comments in the text**

    **line**       *reviewer comment*

              our response

              *adjustments in the text*

    **20ff**       *Number of samples: In line 20 you speak about 400 samples from 10 species, and in*

              *line 22 'six of these taxa', it not clear how many samples (from those 400) were*

*investigated for n-alkanes.*

We add the sample number of *n*-alkane measurements in the sentence.

*Additionally, n-alkane patterns of six of these species (n = 16) were investigated.*

**Table A2** *why is there so much variation in the number of samples analysed (n)? n for 15N is lacking*

As we mentioned in the Methods, the C/N ratio of mosses has a wide range. Thus, the high weight needed for the sample replicates for the $\delta^{15}N$ measurements meant that in some cases there was not enough material to make this measurement. Furthermore, n for $\delta^{15}N$ is not lacking, but it seems that the table format was not optimal, so we 'left aligned' the headings and shaded the columns.

The sentence of the isotope methods was added as follows:

*Due to the relatively wide range of C/N ratios of mosses, we used 1.5 mg for each carbon stable isotope measurement and a replicate of 3.0 mg for each nitrogen stable isotope measurement and the analysis of elemental composition. The high weight needed for the nitrogen sample replicates prevented the measurement of $\delta^{15}N$ and the C/N analysis for some samples.*

**35** *is '. . . as a function of microbial symbiosis' also an assumption? If not from which data is this concluded?*

It is an assumption. We refine the sentences as follows:

*We find differences in $\delta^{13}C$ and $\delta^{15}N$ signatures between both habitat types. For some species of the meso-hygrophilic group, we suggest that a relationship between the individual habitat water-level and isotopic signature can be inferred as a function of microbial symbiosis.*

**37** *...) and intermediate (C25) chain length, respectively.'*

We accept the reviewers comment and added the intermediate chain length:

*[...]of n-alkanes with long (n-$C_{29}$, n-$C_{31}$) and intermediate (n-$C_{25}$) chain lengths, respectively.*

**37 - 39** *Be more specific. 'Isotopic and biochemical signals' are mentioned, I recommend using C/N and n-alkanes instead of biochemical. Also 'certain moss taxa' could you mention them?*
*I would be more careful in exptrapolating the results to be useful in paleaoenvironmental studies, this is not the subject of the ms, [...]*

We accept the comments and changed the sentence as follows:

*Overall, our results reveal that C/N$_{(wt)}$ ratios, isotopic signals and n-alkanes of studied brown moss taxa from polygonal wetlands are characteristic of their habitat.*

**49** *Do you mean mosses in general with 'their'? the abundance of mosses in peat and permafrost is probably more related to ecology (cold wet conditions), instead of to its intrinsic low decomposition rate.*

Yes we agree with the reviewer, but we mention the advantages for mosses and the 'surface' ecology in the sentence before. For example, we show in Zibulski et al. 2016 the recent vegetation composition on the surface and the dominance of mosses. The commented sentence focused on the buried organic material, which shows, in contrast to recent vegetation composition, a dominance of mosses. Products of their second metabolic path protect moss material from degradation by fungi and microbial organisms for example.

**51 - 52**   *I don't think that little is known about stable isotopes and 'biochemical characteristics' of mosses, the effect of this study (habitat) is very interesting and indeed not much of it is known. In the next sentence (line 56-58) you say that these are the most commonly used parameters, isnt'that contradicting?*

*I recommend to use 'n-alkane and C/N ratio' instead of 'biochemical characteristics' throughout the paper. Because the term biochemical characteristics is much broader*

In comparison with vascular plants, little is known. We accept the reviewers comment and refined the sentence to dispel misunderstandings with 'the most commonly used parameters'.

*Despite the significance of mosses in high-latitude biodiversity and matter cycles only little is known about their C/N ratio, stable isotopes and n-alkane characteristics in comparison to vascular plants.*

**60**   *microbial material instead of activity?*

We used the reference of Chanway et al. (2014) and they used 'microbial activity'. Furthermore, the microbial activity in such regions depends on a lot of processes and is an important parameter of decomposition.

**75**   *Explicitly mention this point earlier (already in the Abstract), because mosses and peatlands are often associated with Sphagnum. Indeed this study is more novel especially of the focus on non-sphagnum mosses, but this is not clear from the beginning and may also solve the other problem mentioned above.*

We accept and changed the term 'mosses' to 'brown mosses' from the beginning.

**104**   *This is a very good idea, I would mention this already in the Abstract, see earlier comments. But then I see the Methodology in lines 124-131 and the question arises if this really has been studied because mean values are used to determine its hydrological habitat, thereby loosing important information on the effect of water level. Relevant information is missing on how many plants were studied for each of these means and how is the variation within each group?? It is again not clear to me how the the two groups (xero-meso and meso-hygro) are defined, on the basis of species or hydrological habitat?*

Due to the unknown 'real plant position in relation to water-level', we think the effects of water-level are expressed in the single representations of results for each individual species with their specific habitat requirements.

**135**   *abbreviations have already been used*

We have therefore corrected the sentence as follows:

*The total content of carbon and nitrogen and the ratio of stable isotopes were measured with a DELTAplusXL isotope ratio mass spectrometer [...]*

**178**   *perhaps individual instead of single?*

We have changed the sentence as follows:

*The medians of the individual species in the xero-mesophilic group [...]*

| 188 | *Fig. 3b?* |
| --- | --- |
| | Thank you for this comment, we added the Figure reference. |
| | *[...] exhibit a positive relationship between $\delta^{15}N$ values and position relative to the water-level (Fig. 3b).* |

| 190 | ***If the alkanes are meant to use as a proxy for species or water level, then the data of absolute abundance must be presented as well, as this is highly variable like you indicate here. In a mixture of species like in peat and soil, a certain distribution can only be related to the species (or water level) if differences in absolute abundance is taken into account.*** |
| --- | --- |
| | We agree with the reviewer's comment and have added the absolute abundances of $n$-alkanes (Table 2). Furthermore, we calculated the $ACL_{21-33}$ and the $P_{aq}$ for comparison to other data and complete the Discussion section '4.4 $n$-alkane patterns'. |

| 195 | ***I don't understand this. According to the figure there is not really a species- preferred position all a-d have a -3 and 10 for example. If you look at the highly variable water levels within each species in Fig. 4, then I would not say that the species can be separated into those groups of xero and mesophilic. It must be more explicitly mentioned how these groups are defined, based on what.*** |
| --- | --- |
| | Due to the added Table 1 and our explanations how we determine the mean plant position in relation to water-level in the Methods section 'Sampling and studied moss species', we think this misunderstanding is dispelled. |
| | We refined the caption of Fig. 4 as follows: *The relative amounts [%] of n-alkanes with an odd number of carbon atoms in selected brown mosses grouped by species and ordered by their preferences with respect to their mean plant position relative to water-level for xero-mesophilic (a, b) and meso-hygrophilic (c, d) mosses.* |

| 196/197 | ***I would delete the enter between lines*** |
| --- | --- |
| | We deleted the line return between both lines. |

| 225-226 | ***Now I understand it better. This is a pity. It would be very interesting do such a study, correlate the water level to the plant chemistry, at the plant detail, not the plot. It should be made very clear in the Method Section, now I had the impression that it was done on the individual plant level.*** |
| --- | --- |
| | We are very sorry for the misunderstanding; we have refined the method part to clarify the explanation for the calculation of the mean plant position in relation to water-level. |

| 228-229 | ***this must be mentioned must earlier.*** |
| --- | --- |
| | Yes, we now use 'brown mosses' throughout to make it clearer. |

| 297 | ***but there is a large variability. See line 195*** |
| --- | --- |
| | Correct, but we do not think it is relevant here. |

| 329 | ***I do not understand the word choice 'individuals' in the context of the sampling design, see earlier comments*** |
| --- | --- |
| | We agree with the reviewers comment and changed the word. |
| | *We also find that n-alkane patterns of recent brown mosses are species' specific characteristics, with only minor modifications imposed by the hydrological conditions.* |

**333**     *no fossil material was studied, so this conclusion cannot be made*

We accept the comment and deleted this conclusion.

[revised manuscript text omitted]
_{21-33}$) for *Sphagnum* is reflected by our measurements for *Drepanocladus* and *Scorpidium*, i.e. they show a dependence with water level. As we observed a clear difference in the $ACL_{21-33}$ between the xero-mesophilic and the meso-hygrophilic group, we suggest that the inclusion of mid-chain $n$-alkanes ($n$-$C_{21}$ to $n$-$C_{25}$) in the equation of ACL improves its value as a proxy for moisture conditions. Andersson et al. (2011) inferred $ACL_{27-31}$ values of 29 for brown-moss peat from western Russian during wet phases, which is however, poorly comparable to our results because they investigated total peat organic matter instead of pure moss material.

Ficken et al. (2000) proposed $P_{aq}$ as a semi-quantitative proxy ratio for the differentiation of terrestrial and aquatic plants (<0.1 terrestrial plants, 0.1–0.4 emergent macrophytes, 0.4-1 submerged/floating macrophytes). Our inferred $P_{aq}$ results for the individual species fit well with these assumptions. For example, submerged/floating *Scorpidium* ($P_{aq}$ median: 0.56) has a higher median $P_{aq}$ value than floating/mesic *Drepanocladus* ($P_{aq}$ median: 0.34), which is again higher than that of the xero-mesophilic mosses *Tomentypnum* ($P_{aq}$ median: 0.13) and *Aulacomnium* ($P_{aq}$ median: 0.13). Ficken et al. (2000) also measured a *Drepanocladus* sample ($P_{aq}$ = 0.30). 
[revised manuscript text omitted]

---

## Author Response (AR1)

*Reply to editors' comments concerning the manuscript:*

**"C/N ratio, stable isotope ($\delta^{13}C$, $\delta^{15}N$), and n-alkane patterns of bryophytes along hydrological gradients of low-centred polygons of the Siberian Arctic"**
by R. Zibulski et al.

*Received 16$^{th}$ December 2016*

We thank editor for the valuable comments which have contributed to the improvement of the article.

**General comments and comments in the text**

| | |
|---|---|
| **line** | *reviewer comment* |
| | our response |
| | *adjustments in the text* |

*1. Regarding ratio C to N. The reviewer did not mean necessarily that you added the word "by weight". I would rather change it to molar ratio to make it comparable con literature. For instance (Meyers and Ishiwatari 1993, used (as it was described originally) molar ratio.*
We calculated the molar ratios, and we changed the text and figures 2, 3a, and 3b.

*2. Abstract. Sentence "…ratios (by weight), δ13C and δ15N data of 400 brown …[SEP]" is misleading since there is data for less than 20 samples not 400. Please change accordingly.*
We describe it correctly. There were 400 samples for isotopic and elementary measurements (as we described in the information about individual *n* for each species in table A2) and only 16 samples for the *n*-alkane study.
We added the sample counts for each measurement and changed the method part as follows:
*Due to the relatively wide range of C/N ratios of mosses, we used about 1.5 mg for each carbon stable isotope measurement (n = 400) and a replicate of 3 mg for each nitrogen stable isotope measurement (n = 326) and the analysis of elemental composition.*
*n-Alkane analyses were performed on a subset of 16 samples.*

**120**    *3. Regarding word "preferred". Is it proven that there is preference? Or just they are there? Please change accordingly.*
Plant distribution depends on water-level and the importance of micro relief for bryophyte species distribution was demonstrated in the paper "Vegetation patterns along micro-relief and vegetation type transects in polygonal landscapes of the Siberian Arctic" (Zibulski et al., 2016). The studied species show an azonal distribution depending on water-level gradient (micro relief) within low-centred polygons. Furthermore, we calculated the mean plant height for this dataset, which matches with our field observations and the descriptions in the relevant literature for these well-known species and thus we consider the use of "preferred" acceptable in this instance.

**123-124**    *4. "Furthermore, because of the stability provided by water, they do not need to invest in a sturdy stem-structure and accordingly have lower C contents in their biomass." In an abstract, you convey the message of your results; you did not study sturdiness of mosses,*

*Did you?*
That is correct, we have now deleted this assumption and the relevant sentences.

***5. I understand the term "signature" for d13C, and d15N, but you really mean isotopic composition. I don´t see why not use proper terms.***
We accept the editor's comment and changed the term from signature to composition in the whole manuscript.

***6. Abstract, Line 119. Your statistic results in significant differences, but it appears that there are large standard deviations (Fig. 3) and not much differences are evidence in C/N. Are you sure that all requirements for a t-test are fulfilled? Normality, homoscedasticity, etc. C/N of Hylocomium splendens ( xero-mesophyloic) and of Hamatocaulis lapponicus (meso hygrophylic) in Fig. 3 are essentially the same but different according to the statistics. What is the value of this information to make it to the abstract ("main findings")? Please change accordingly.***

The value of this information is the difference between moisture conditions. Some of the species with large standard deviations show an expected intraspecific variability between the measured parameter and water-level.
We have checked the $C/N_{(m)}$ data against normality giving the following result:

[Figure]

The xero qq-plot looks weak for normality, and the sharp Shapiro-test for this group (`W = 0.97474, p-value = 0.02839`) failed the t-test for normality, but the t-test is robust against infringement of the requirements for data with n>30 ($n_{(xm)}$ = 152).

Additionally, we tested the homoscedasticity of the data with the following results:
```
Levene's Test
       Df F value Pr(>F)
group   1  0.6606 0.4169

Two Sample t-test
t = 8.1707, p-value = 6.841e-15
```

Also a Mann-Whitney-U test, to ascertain the differentiation of both groups, gave a significant result for the C/N ratio:
`W = 18280, p-value = 1.83e-13`

We can conclude that both groups are non-identical ($\alpha$=0.05).

**+++++++++++++ d$^{13}$C**

Check for normality of both groups:

[Figure]

```
Shapiro-Wilk normality test
W = 0.97466, p-value = 0.0002813 → failed
```

Homoscedasticity for d$^{13}$C is also not evidenced:
```
Levene's Test
      Df F value   Pr(>F)
group  1   52.03 2.78e-12 *** → failed
```

Due to the heteroscedasticity of d$^{13}$C data and the failed normality, we used the Mann-Whitney-U test
```
W = 16232, p-value = 0.007988  → new result
```

We can conclude that both groups are non-identical (α=0.05).

**+++++++++++++ d$^{15}$N**

Tests of Normality:

[Figure]

```
Shapiro-Wilk normality test
W(xm) = 0.99295, p-value = 0.8285
W(mh) = 0.99304, p-value = 0.4156
```

Homoscedasticity

```
Levene's Test
      Df F value Pr(>F)
group  1  0.0135 0.9076

Two Sample t-test
t = -6.9599p-value = 1.878e-11
mean in group 1 mean in group 2
     -2.869826        -1.664576
```

We can conclude that both groups are non-identical ($\alpha$=0.05) in their d$^{15}$N values.

We refined our statements and changed the figures and the different parts as follows:

Abstract: *Overall, we find high variability in all investigated parameters for two different moisture-related groups of moss species.[...] The C/N$_{(m)}$ ratios range between 11 and 53 (median: 32) and show large variations at the intraspecific level.*

Results: *However, the medians of the C/N$_{(m)}$ values of the xero-mesophilic species ranging from 47.6 to 52.9 (Fig. 2) are significantly higher than those of the meso-hygrophilic group, which range from 37.1 to 46.5 (W = 18280, p << 0.001). The C/N$_{(m)}$ ratios show no intraspecific relations among individual species and water-level (Fig. 3a), except for Tomentypnum nitens (r² = 0.11, p < 0.05).*
*The medians of the individual species in the xero-mesophilic group (range: -29.4 to -27.1‰) are significantly different (W = 16232, p = 0.008) from those of the meso-hygrophilic group (range: -29.2 to -24.8‰).*

Discussion: *Our results reveal that averaged C/N$_{(m)}$ ratios for the xero-mesophilic moss group are higher than for the meso-hygrophilic group, probably reflecting the known difference between terrestrial and aquatic plants (Meyers and Ishiwatari, 1993). There are two possible impacts, which can influence the C/N ratio of these groups: (1) competition with vascular plants and (2) accessibility of nitrogen pools . [...]. The large variability in the C/N data may be a result of atmospheric conditions and organic matter degradation being the principal sources at xeric sites, whereas in mesic and wet sites microbial symbionts play an important role in the C/N ratio. However, the signal-to-noise ratio is probably too low to give a meaningful result because only the average water level of each plot but not of each individual plant was recorded.*

*[...] individuals growing at dry sites showed higher medial $\delta^{13}$C values than those growing at wet sites. A difference among the two habitat groups is observed; they partly contradict the intraspecific findings in that some of the xero-mesophilic species known to prefer dry rims such as Hylocomium splendens and Tomentypnum nitens have particularly low $\delta^{13}$C medians. [...]The detected differences in moss $\delta^{13}$C values, particularly of the meso-hygrophilic group, either reflect a source signal depending on water-level or a physiological reaction of the plant related to water-level [...]*

*Thus, the large ranges within several species of meso-hgryophilic habitats in arctic regions suggest that the existence of open water leads to more depleted $\delta^{13}$C values and measurements of the isotopic composition of methane when present and microbial groups in the water and terrestrial litter should be possible.*

Conclusions: *With respect to the isotopic source pools, the meso-hygrophilic species have greater access than xero-mesophilic species, which is seen in their large ranges. The approximate habitat-specific division of $\delta^{13}$C values as a result of discrimination by RuBisCO under different hydrological regimes is overturned by the influence of different*

*sources and cannot provide a clear distinction from a single measurement of either habitat type.*

**124-129**   *7. With the exception of two cases (Fig. 3b), most determination coefficients (regression) are lower than ca. 10% or non-existent; and you suggest "… that a relationship between the individual habitat water-level and ⌷isotopic signature wascan be inferred as a function of microbial symbiosis ". I do not see evidence for that. Please change accordingly.*

Yes we agree with the editor and have deleted this assumption.

*8. Was sampling in 2011 and 1012 during the same season? Should this source of variability be considered? Why?*

We accept the editor's comment and describe the vegetation season in the methods (2.1 Sites) more precisely. We chose July and August for our expeditions to avoid the influence of different water sources during spring flooding and because of the low impact of environmental parameters such as absence of circadian rhythm cycle and small temperature variations during a day in the summer north of the polar circle.

We added the sentences as follows:

*The plant material was collected during the vegetation season (July–August) from eight low-centred polygons located along a zonal vegetation gradient ranging from open forest via the forest-tundra intersection to subarctic tundra (Matveev, 1989) to obtain a representative sample set of northern Siberian lowlands (Fig. 1).*

**327-329**   *9. How can a general trend can be deduced if 17% of variance of first 2 axes?*

We accept the editor's comment and refined the sentence:

*The observed trend was also assumed in the biplot of the first two PCA axes, even though their declared variance is relatively low (16.9%) in the dataset.*

*10. Regarding Paq, ACL, and CPI. ⌷What is the conclusion of this comparison? Can they be graphically compared? Do they support your conclusions? Reviewer 2 specifically ask for inclusion of those indices since "n-alkane data are inadequately presented and interpreted"*

The ratios of $ACL_{21-33}$ and $P_{aq}$ support the differentiation between both environmental groups. We created a figure of these data, but we think it is unnecessary and inflates the manuscript. We abstain from a mathematical evaluation and interpretation of ratios within both environmental groups due to the small *n* for most of the individual species. Thus, we still show the ratios in table form (Table 2) and supplement the text in the Results and Discussion sections with a refined presentation and interpretation.

[Figure]

Discussion:
ACL21-33:

> **dry** (ACL$_{21-33}$ = 29.1 – 27.5)   **< moisture condition >**   (ACL$_{21-33}$ = 26.4 – 25.2) **wet**
> Tom_nit   <   Aul_tur   <   Aul_pal   <   Dre_rev   <   Sco_sco

Paq: *Our inferred P$_{aq}$ results for the individual species agree with these assumptions. If we consider that the proxy ratio levels were created by vascular plants from a limited dataset of lakes in Kenya and as we focus on non-vascular plants of the arctic, we chose other level terms.*

| *terms by Ficken et al. (2000)* | *emergent macrophytes* | *submerged/floa...* |
|---|---|---|
| | | *macrophytes* |
| *adapted terms for mosses* | *xero-mesophilic mosses* | *meso-hygrophilic ...* |
| *species sorted by P$_{aq}$*   Aul_tur | <   *Tom_nit*   <   *Aul_pal* | <   *Dre_rev*   <   ...* |

Conclusions: *The applicability of proxy ratios (ACL$_{21-33}$ and P$_{aq}$) could be attested for arctic mosses after adjustments of the levels.*

**11. "The total content ⌈SEP⌋ of carbon". Total is redundant since we hope you are measuring total and not part of it.**
We accept the editor's comment and delete "total".
*The content of carbon and nitrogen and the ratio of stable isotopes were measured [...]*

**12. The argument "Considering the low n-alkane ⌈SEP⌋… " is not necessary since we always have to use quantification standard independent of low or high contents**
We have changed the sentence as follows:
*Five µg of the quantification standard (5α-androstane, 1-ethylpyrene, 5 α-androstan-17-one and erucic acid) were added.*

**13. Replace calculate with calculated**
Done.

*Additionally, we calculated the average chain length [...]*

**298**     ***14. R2 is regression, not correlation. What do you want to detect? 2 dependent variables or one dependent and one independent?***

We calculated the regression because plant position in relation to water-table is an independent variable and the C/N ratio, $d^{13}C$ and $d^{15}N$ are dependent variables. We replace "correlated" with "related" in the text.

**257**
**264**     ***15. -some samples. Specify how many? Same with line 264. Does it change your conclusions?***

There were 74 samples from which we could only measure $C/d^{13}C$. Due to the addition of n in the previous sentence, it does not change our conclusion, because there were still more than 300 samples for N and from both groups (xero/meso).

We added in the sentence before: *Due to the relatively wide range of C/N ratios of mosses, we used about 1.5 mg for each carbon stable isotope measurement (**n = 400**) and a replicate of 3 mg for each nitrogen stable isotope measurement (**n = 326**) and the analysis of elemental composition.*

**340**     ***16. Xeric and mesic [SEP] are adjectives and need a noun such as environments, or soils***

We choose as noun 'growing conditions'. "xeric and mesic" refer to "conditions".

**334-**
**363**     ***17. See comment # 6***

We changed the sections see comment #6

**475**     ***18. " clear differences in ACL" are not so clear in the text. Please change accordingly.***

We present our data in Table 2 and support the clear difference with this little series in the discussion.

**dry** ($ACL_{21-33}$ = 29.1 – 27.5)     **< moisture condition >**     ($ACL_{21-33}$ = 26.4 – 25.2) **wet**

Tom_nit    <    Aul_tur    <    Aul_pal    <    Dre_rev    <    Sco_sco

**530-**
**532**     ***19. See previous comments on C/N and d13C***

We do not follow you, because lines 530-532 are citations, which were added in our previous reply to the reviewer comments. Despite searching for other possibilities, we could not see what problem your comment referred to.

***20. Reviewer 1. # 35, about symbiosis. See my comment # 4***

Yes, this is right, but as this is the Discussion section we treat it is a point of interest that should be considered for following measurements.

We deleted this assumption from the Abstract and Conclusions.

[revised manuscript text omitted]